Corrected: Author correction

# Uplift of the central transantarctic mountains

Phil Wannamaker [1], Graham Hill[2,3,4], John Stodt[5], Virginie Maris[1], Yasuo Ogawa [6], Kate Selway[7], Goran Boren[8], Edward Bertrand[9], Daniel Uhlmann[10], Bridget Ayling[11], A. Marie Green[12] & Daniel Feucht[13]

The Transantarctic Mountains (TAM) are the world's longest rift shoulder but the source of their high elevation is enigmatic. To discriminate the importance of mechanical vs. thermal sources of support, a 550 km-long transect of magnetotelluric geophysical soundings spanning the central TAM was acquired. These data reveal a lithosphere of high electrical resistivity to at least 150 km depth, implying a cold stable state well into the upper mantle. Here we find that the central TAM most likely are elevated by a non-thermal, flexural cantilever mechanism which is perhaps the most clearly expressed example anywhere. West Antarctica in this region exhibits a low resistivity, moderately hydrated asthenosphere, and concentrated extension (rift necking) near the central TAM range front but with negligible thermal encroachment into the TAM. Broader scale heat flow of east-central West Antarctica appears moderate, on the order of 60–70 mW m$^{-2}$, lower than that of the U.S. Great Basin.

[1] University of Utah/Energy & Geoscience Institute, Salt Lake City, 84108 UT, USA. [2] University of Canterbury, Gateway Antarctica, Christchurch 8041, New Zealand. [3] Antarctica Scientific Ltd, Wellington 6011, New Zealand. [4] Formerly at GNS Science, Natural Hazards Division, Lower Hutt 5011, New Zealand. [5] Numeric Resources LLC, Salt Lake City, 84108 UT, USA. [6] Tokyo Institute of Technology, Volcanic Fluid Research Center, Tokyo 152-8550, Japan. [7] Department of Earth and Planetary Sciences, Macquarie University, Sydney, 2109 NSW, Australia. [8] Department of Geology & Geophysics, University of Adelaide, Adelaide, 5005 SA, Australia. [9] GNS Science, Natural Hazards Division, Lower Hutt 5011, New Zealand. [10] First Light Mountain Guides, Chamonix 74400, France. [11] Great Basin Center for Geothermal Energy, University of Nevada, Reno, 89557 NV, USA. [12] Department of Geology & Geophysics, University of Utah, Salt Lake City, 84112 UT, USA. [13] Department of Geological Sciences, University of Colorado, Boulder, 80309 CO, USA. Correspondence and requests for materials should be addressed to P.W. (email: pewanna@egi.utah.edu)

Compressional mountain belts such as the Himalaya or the Andes are supported by massive crustal roots in Airy compensation[1], [2]. Extensional mountain belts are more mysterious, and several mechanisms are suspected of operating beneath the high shoulders that bound rift domains. A leading interpretation for uplift of interior extensional orogenies, for example, the western U.S. and Africa, has been replacement of lithospheric upper mantle with hotter asthenosphere, which is less dense due to thermal and possibly melting-induced density reduction[1], [3–8]. Nevertheless, the widespread presumption of a thermal cause for rift uplift is not proven, and thermal activity potentially could obscure other causes such as flexural unloading or solid-state density contrasts. The central Transantarctic Mountains (TAM) are the core of the world's largest rift shoulder[9], and constitute an ideal testbed for illustrating the primary mechanisms that may be at play in rift dynamics. They are nearly 3500 km-long with relief exceeding 4 km. The TAM are the fundamental divide of the Antarctic continent, and a key element in plate tectonic reconstructions, climate history, and biological evolution worldwide.

East Antarctica is a stable, Precambrian cratonic shield that previously formed part of Rodinia and Gondwana super-continents[10]. Its western region including our survey area may represent a conjugate paleo-rifted margin to modern south-western Laurentia. WA was assembled against EA from accreted blocks and subduction-related plutonism (e.g., Cambrian Ross Orogeny) from early Paleozoic until middle Mesozoic time[11–13]. The primary extension phase is tied to Gondwana breakup and began in Jurassic time, migrating into the WA system in the Cretaceous[14]. Secondary rifting from the middle Cenozoic until today is strongly transtensional and concentrated adjacent to the TAM and Marie Byrd Land[15–18]. Most TAM exhumation and

uplift began then (55–45 Ma), suggesting establishment of a regional boundary fault separating WA and EA[11], [14]. In the Shackleton Glacier area near to the grid northwest of our study, uplift and denudation of 4.9–6 km was focused in the 40–14 Ma time frame with <1 km of uplift since[15]. The concentration of extension and subsidence-related sediment buildup close to the TAM near Ross Island is consistent with the later-stage (middle to late Cenozoic), lithospheric rift necking[19].

Early seismic surface-wave studies demonstrated the distinction between EA and WA, with EA showing craton-like velocities to >200 km depth, whereas WA shows reduced upper-mantle velocities at depths of ~100 km[20], [21]. These continental-scale models exhibit a strong velocity gradient along the TAM but have lateral resolution generally coarser than ~100 km due to the sparse seismometer coverage. Significant improvement has resulted from more recent deployments, especially locally with targeted transects and networks[22–24]. Substantial resolution gains have been in Marie Byrd Land to Ellsworth-Whitmore Mountains sector, and toward the Gamburtsev Mountains, from the POLENET, TAMSEIS, and GAMSEIS arrays[22], [23]. Focused study from Ross Island through North Victoria Land, where late Cenozoic volcanism is pronounced, shows low mantle velocities mainly in the upper 300 km, suggesting that rift-related or local convection processes are likelier than deep-seated plume upwelling as the thermal driver[22], [24]. However, the central TAM range front that we roughly define as the segment spanning Byrd Glacier through Shackleton Glacier[9], remains coarsely sampled seismically.

To discriminate processes underpinning the TAM and reveal the architecture of the rift transition from West Antarctica (WA) to East Antarctica (EA), we acquired a transect of magnetotelluric (MT) geophysical soundings about 550 km in length through the

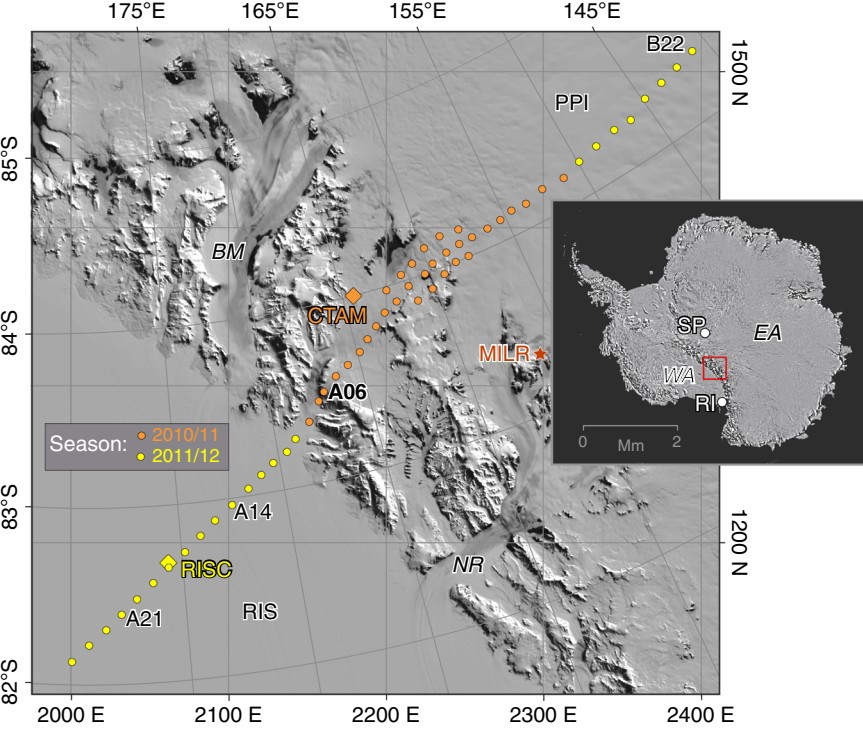

**Fig. 1** MT site locations across the central TAM and surroundings. Locations and physical features are plotted using a MODIS satellite base map[72]. Grid north is up and both latitude–longitude and Universal Polar Stereographic (UPS) coordinates are overlain on main panel. Orange diamond denotes CTAM multi-investigator camp location active during 2010–11 austral summer season (orange sites), whereas yellow diamond denotes Ross Ice Shelf field camp (RISC) for 2011–12 season (yellow sites). Specific sites discussed in text are labeled. The sole passive seismic station[22], [23] within the field of view is MILR. Beardmore and Nimrod glaciers are BM and NR, whereas RIS and PPl are Ross Ice Shelf and Polar Plateau, respectively. Within the TAM, the MT transect lies along the lesser Law–Walcott–Lennox–King glaciers. In the inset, WA and West Antarctica, EA is East Antarctica, SP is South Pole, and RI is Ross Island

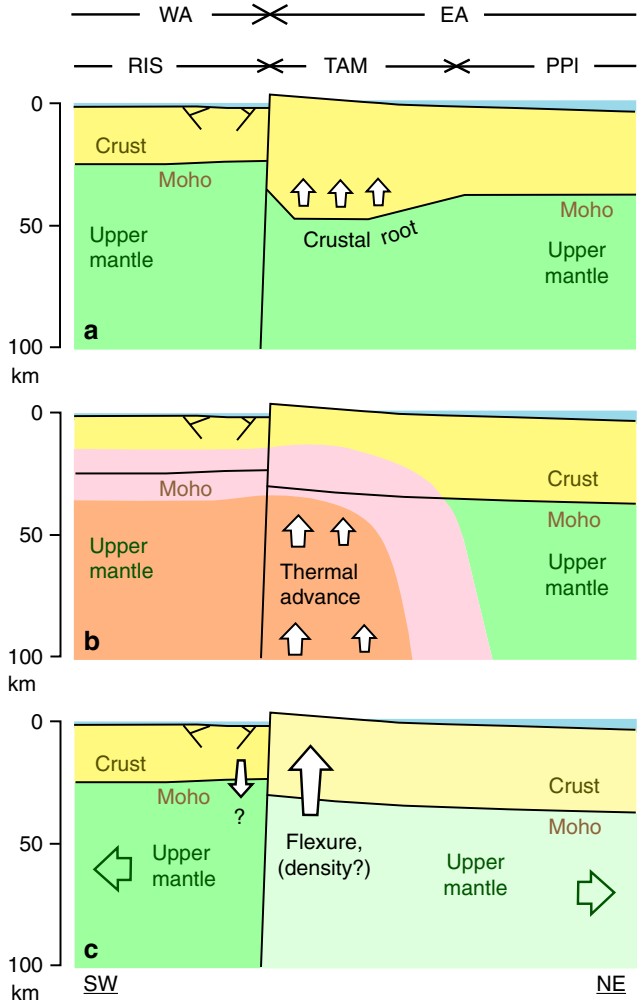

**Fig. 2** Hypothetical uplift mechanisms for TAM rift shoulder. These are **a** buoyant uplift via low-density crustal root; **b** uplift via lateral heating, thermal expansion, and possible melting; **c** uplift via other mechanisms such as lithospheric cantilevered flexure with or without regional density contrasts. Physiographic regions include West Antarctica (WA), East Antarctica (EA), Ross Ice Shelf (RIS), Transantarctic Mountains (TAM), and Polar Plateau (PPI). Diagram not to scale

central TAM (Fig. 1). With MT, images of subsurface electrical resistivity (or its inverse, conductivity) can be produced using natural electromagnetic (EM) wave fields[25]. Resistivity in turn is sensitive to temperature, melts, and fluids. MT surveys have clarified extensional processes elsewhere worldwide in terms of thermal state, rheology, and cryptic tectonism[1, 4, 26, 27]. We show that the central TAM indeed are held up by a non-thermal mechanism, most likely cantilevered flexure, in contrast to more conventional views.

## Results

**Testable alternatives for TAM geodynamics.** Simplified concepts of the region have considered EA as an elastic plate on a fluid substratum (asthenosphere) with a free-edge paralleling the TAM rangefront[28, 29]. Hypotheses for uplift of the TAM and the nature of the WA–EA tectonic transition fall into three main categories (Fig. 2). We are able to evaluate these using the results of our MT surveying across the central TAM integrated with other geophysical information.

First, a crustal root below the TAM, suggested to be residual from regional extensional collapse of West Antarctica, could contribute to TAM uplift through Airy compensation, perhaps in concert with other mechanisms[30–32] (Fig. 2a). However, recent seismic receiver function data near Ross Island indicate that a crustal root near the range front should be no more than 2–3 km and may be entirely absent[33]. According to airborne and GRACE satellite gravity observations, a possible TAM root tangentially proximal to the South Pole may exist in contrasting Neoproterozoic basement[31, 32]. However, the GRACE data modeling implies that any such root has diminished to within estimation error (±3 km) grid southeastward to our study area[32]. Thus, Airy isostasy, likely, contributes negligibly below the central TAM. MT data alone do not confirm or negate the existence of a crustal root because dry or unmelted silicate lower crust is generally resistive regardless of composition[25]. Nevertheless, independently demonstrating through seismic or gravity data, the inadequacy of root buoyancy to explain uplift strengthens the overall ability of our MT results to rank geodynamic mechanisms operating beneath the central TAM.

Second, TAM uplift processes have been compared with those of the margins of the extensional U.S. Great Basin, in particular, lithospheric replacement by hot asthenosphere of lower density[16, 34] (Fig. 2b). Numerous models of TAM uplift near Ross Island have incorporated thermal buoyancy as an essential uplift load in addition to erosion, ice load, and flexural rigidity. Thermal forces considered include lithospheric-scale simple shear with possible magmatic underplating[35, 36], quasi-conductive warming from WA[28, 29], asthenospheric upwelling from small-scale convection associated with Ross Sea margin rifting[22–24, 37], and enhanced lower crustal heat production below the TAM[38]. However, these studies focused on the Northern Victoria Land/Ross Island region where upper mantle is anomalously slow, whereas the upper-mantle state farther toward our study area is less well understood. MT observations are sensitive to the thermal state of the deep crust and upper mantle; moderate mantle resistivities compatible with an average adiabatic upper-mantle geotherm and deep crustal low resistivities indicative of magmatic underplating and fluid release are characteristic of both Great Basin rift flanks[1, 4, 39].

Third, uplift of 5 km or more near the TAM range front on the EA side with appropriate lateral wavelength (~500 km) can be simulated primarily via Vening–Meinesz style cantilevered flexure involving the regional boundary fault, taken to be steeply grid southwest-dipping[40] (Fig. 2c). In this model, the WA and EA lithospheres are decoupled across the dipping boundary fault. Surface deformation is produced through isostatic response to movement along the fault without application of other vertical loads. In particular, the footwall (EA) rebounds isostatically with curvature determined by its effective elastic thickness Te. The currently inferred value near 100 km for the Te of EA[28, 29] would be reduced substantially according to flexural modeling if either sub-TAM radiogenic heat sources were present or lateral warming of TAM lithosphere by thermally active WA had occured[40]. Curiously, a companion flexural rift trough on the near WA side is not observed in the SERIS seismic reflection data at the TAM front ~100 km to the southeast of our profile[41], implying a more complex deformation history for the hanging wall. In addition, the greatly differing histories of WA and EA lithospheres suggest that large-scale, chemical composition-based density contrasts may also exist. If flexure is the dominant TAM uplift mechanism, high electrical resistivities should underlie it close to the range front and well into the upper mantle. Our MT data will allow us to test whether non-thermal uplift causes should instead be preferred over thermal ones in modeling TAM geodynamics.

A hypothesis that we specifically do not consider is TAM rift shoulder isostatic uplift resulting from pure shear extensional necking of WA lithosphere[42]. First, the timing is problematic as

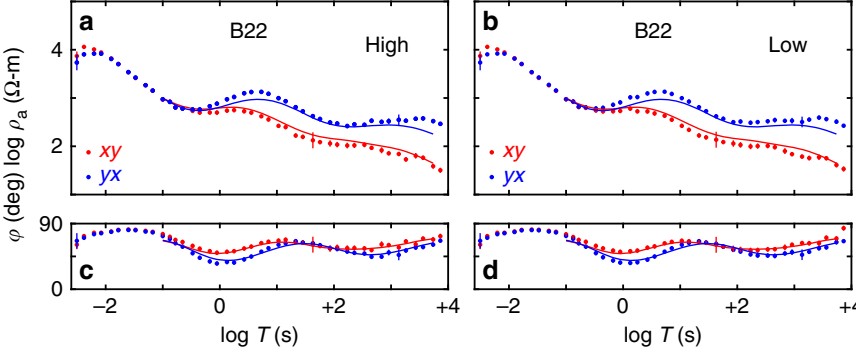

**Fig. 3** Example MT responses CTAM survey area. Shown are off-diagonal apparent resistivities $\rho_{xy}$ and $\rho_{yx}$ (**a**, **b**) and impedance phases $\varphi_{xy}$ and $\varphi_{yx}$ (**c**, **d**) as a function of period $T$ at polar plateau (PPl) site B22 (Fig. 1) corresponding to high- (left) and low- (right) activity times of the diurnal geomagnetic variation signal during the seven day recording interval of January 11 through January 18, 2012. The error floors assigned in the non-linear 3D inversion are reflected in the error bars, and are similar to the data symbol heights. The computed $\rho_a$ and $\varphi$ responses from the inversion model of Fig. 6 are plotted as solid lines

the great majority of WA extension occurred well before TAM uplift[14]. Second, the deep levels of necking required to simulate TAM uplift are modeled to produce smooth Moho topology and large negative gravity anomalies on the Ross Sea side, unlike what is observed[36].

**MT observations across the central TAM.** Fifty-seven MT soundings were acquired over two field seasons (Fig. 1), described further in "Methods". With the MT technique, in order to normalize against variations in the natural source current strength, the observed electric ($E$) and magnetic ($H$) field components in the frequency domain are related through complex impedance $Z$ and vertical magnetic 'tipper' $K$ tensors, expanded as[25]

$$E_x = Z_{xx}H_x + Z_{xy}H_y, \qquad (1)$$

$$E_y = Z_{yx}H_x + Z_{yy}H_y, \qquad (2)$$

and

$$H_z = K_{zx}H_x + K_{zy}H_y. \qquad (3)$$

These are the primary MT response functions, solely determined by Earth properties. The measured EM time series contain a broad range of frequencies (or the inverse, wave periods), which are decomposed into the Fourier domain via spectral analysis. Propagation of MT wave fields in the conducting Earth is diffusive; for example, in a uniform half-space, penetration depths are determined by an e-folding distance (skin depth) dependent upon period and resistivity[25].

The $x$ and $y$ coordinate directions of the data can be rotated computationally from the original in-field setup, and for our analysis $x$ is defined as an average assumed strike of the central TAM of grid N315° for all sites and periods. Given the fully 3D analysis we perform, the particular orientation chosen is not critical. Commonly, the $xy$ and $yx$ MT impedance functions are transformed through a simple arithmetic operation to apparent resistivities ($\rho_a$) and impedance phases ($\varphi$) for initial visualization[25]. Converting MT impedance to $\rho_a$ over a uniform half-space returns the resistivity of that half-space, so $\rho_a$ is intended to depict a smoothed version of resistivity structure toward depth as period increases. The phase $\varphi$ typically resembles the slope of $\rho_a$ vs. period and can imply resistivity contrasts or transitions. An example of apparent resistivity and impedance phase sounding for a site on the polar plateau is shown in Fig. 3. However, this conversion is for heuristic discussion purposes only and is still

just a display of observations; not all data characteristics so portrayed can be taken at face value. Rigorous structural estimation requires non-linear inversion (a form of tomography) as discussed shortly.

To counteract the high electrical contact impedance of the snow (firn), custom buffer preamplifiers were placed at the electric field bipole electrodes[43] (Methods and Supplementary Figs. 1–4). Owing to the remote nature of the sources and the high index of refraction of the Earth relative to the air, MT incident wave fields are assumed to be planar in geometry and to propagate vertically into the ground[25]. As our field seasons occurred close to the previous solar activity minimum, EM field amplitudes were low and steady, and non-plane-wave outliers appeared infrequent (Supplementary Figs. 7 and 8). This was verified by robust remote reference processing of the MT sounding data during high- and low-activity intervals of the solar diurnal variation and obtaining nearly identical results (Methods) (Fig. 3).

Site occupation times were in the 4–11 day range yielding good quality MT responses in the wave period range 0.004–3000 s, and sometimes 6000 s, which as we show has allowed us to resolve resistivity through the upper mantle. Our data are richly complex indicating structure at multiple scales (Supplementary Fig. 9). The observations along the main transect are summarized graphically in Fig. 4 as pseudosections, and where the data are contoured vs. position and signal wave period[25]. Despite complexity, the large offset in overall levels of $\rho_a$ that occurs near the boundary between the RIS and the TAM range front is unmistakable, with the TAM and polar plateau (PPl) generally being resistive while the Ross Ice Shelf (RIS) is much less so (Fig. 4a, j). RIS resistivities are influenced by seawater and perhaps Cenozoic sediments under the ice as the low apparent resistivities around 10 s period imply (see Methods and Supplementary Figs. 10–11). However, deeper upper-mantle resistivities will also be shown to be reduced in the resistivity inversion as indicated by the elevated phases at periods >500 s for the RIS sites (Fig. 4). Spatially variable, moderately lower $\rho_a$ toward longer periods over the East Antarctic segment will translate in the inversion to isolated low-resistivity units in the middle crust associated with early lithospheric assembly. A strong positive response in tipper near the RIS-TAM margin at periods in the 30–1000 s range (Fig. 4) in part denotes a compact low-resistivity body of whole-crustal scale.

**3D resistivity of the central TAM.** Because Earth's electrical resistivity can range over orders of magnitude, and propagation of

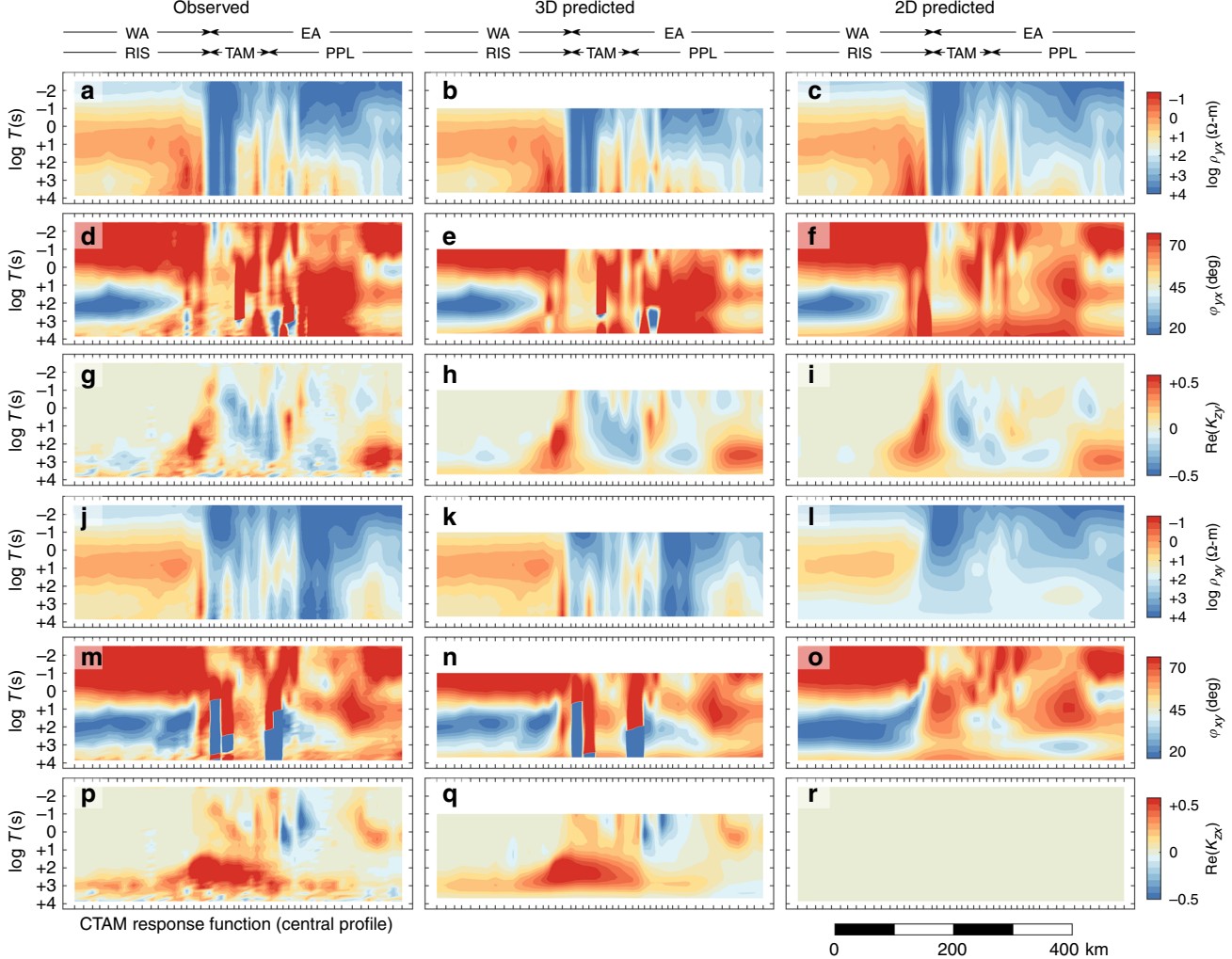

**Fig. 4** Pseudosections of primary MT observations. These are plotted as a function of period ($T$) and distance, both observed and computed from inversion, for stations along the central profile of Fig. 1. These include apparent resistivities $\rho_{xy}$ and $\rho_{yx}$, and impedance phases $\varphi_{xy}$ and $\varphi_{yx}$, from the off-diagonal impedance elements, and the real (in-phase) component of the complex tippers $K_{zx}$ and $K_{zy}$. The x axis for data definition is grid N315°. To ease finite element mesh discretization requirements, the 3D inversion only considered data for $T > 0.1$ s. Included for comparison with the 3D analysis is the calculated response of a purely 2D inversion model described in "Methods" with Supplementary Fig. 14. Physiographic regions are as in Fig. 2. An additional plot of Fig. 4 using a spectral color scheme is presented as Supplementary Fig. 5

MT wave fields there is diffusive, modern resistivity models are generated through non-linear regularized inversion[25]. The regularization typically, and here, is via "minimum-structure", where models are sought that fit the data while simultaneously damping model spatial slope. Given that longer period waves interrogate deeper or farther from the recording sites in diffusive propagation, resolution similarly diminishes with depth or distance[25]. Resolution tests are performed by perturbing model features and assessing change in data misfit. We utilize all four impedance and both tipper elements to minimize the need for a 2D assumption, and allow treatment of a meandering profile plus the patch of soundings near the TAM-PPl transition. The $Z_{xx}$ and $Z_{yy}$ impedance and $K_{zx}$ tipper elements hold directional information and can allow image formation offline even though resolution decreases laterally from the transect.

Our inversion algorithm is based on the finite element method using vertically deformed hexahedra to incorporate topography[44], [45] (Fig. 5) (see "Methods"). The starting model was a 300 ohm-m half-space down to 410 km, underlain by 20 ohm-m in keeping with the resistivity drop expected across the olivine–spinel phase boundary[46]. Models include estimates of site-local impedance

static distortion, which gave modest improvements in data fit (Supplementary Figs. 12–15). The 3D inversion closely reproduces all observed MT components including extreme responses with out-of-quadrant impedance phases (Fig. 4d, e, m, n). An attempt at 2D inversion using subsets of the MT data less influenced by 3D effects was only partially successful (Fig. 4c, f, i, l, o, r), although a qualitatively similar model cross section was obtained (Supplementary Fig. 16). Sensitivity tests (Supplementary Fig. 17) show that the longest periods of our data are sensitive to the 410 km phase transition, thus defining depth of survey penetration.

The best-fitting model (Fig. 6) shows several primary features pertinent to resolving the alternative geodynamic hypotheses of Fig. 2. Foremost, resistivities >1000 ohm-m extend to depths of ~150 km below the TAM range front (feature TL). We tested this feature by imposing conservatively lower resistivities (~300 ohm-m) for ~150 km northeastward under the TAM and demonstrating a significant increase in response misfit for TAM MT sites (Supplementary Figs. 18 and 19). Beyond 200 km depth in Fig. 6, resistivity falls below 100 ohm-m until reaching the low-resistivity, 410 km phase discontinuity. Toward the polar plateau

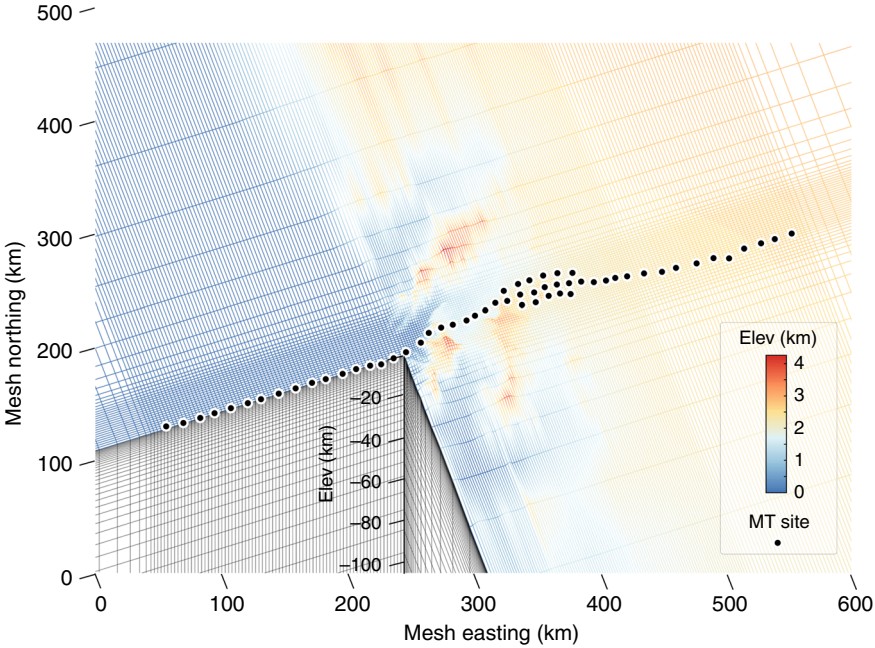

**Fig. 5** Central portion of hexahedral finite element mesh. This mesh is used to simulate and invert MT data across the central Transantarctic Mountains. Element elevations are indicated by color. Each element is a parameter in the regularized data inversion. The mesh coordinates are local, where (0, 0) corresponds to UPS (2006835.5E, 977476N)

in the lower crust, compact low-resistivity bodies are seen including a broad, quasi-horizontal zone beneath the north-easternmost ~150 km of the profile (feature MS). The narrow, grid SW-NE linear behavior of MS is defined by $\rho_{yx}$ decreasing toward longer periods but with $\rho_{xy}$ decreasing only slightly (Fig. 4a, b, j, k). However, the narrow SE-NW arm near its middle is supported by the pronounced crossover in $K_{zy}$ around 1000 s period near the NE end of the profile (Fig. 4g, h). Beneath this, the resistivity rises to >1000 ohm-m again to at least 200 km depth (EA C).

Immediately, grid southwest of the thick resistive TAM domain, a singular low-resistivity zone extends through the adjacent RIS crust into the uppermost mantle (Fig. 6) (feature RN). That this feature appears to grow toward grid southeast (Ross Island direction) follows from the substantial positive magnitude in $K_{zx}$ in the 30–1000 s range at the RIS-TAM boundary (Fig. 4g, h). Removal of this feature also significantly increases misfit of the impedance at sites over that volume (Supplementary Figs. 19 and 20). RIS upper mantle broadly is less resistive than that of the central TAM or PPl, although the upper ~100 km to the southwest approaches 300 ohm-m (WL) (Fig. 6). Deeper toward grid southwest, upper-mantle resistivity falls to near 30 ohm-m (WA A). Although this low resistivity appears to shallow beyond the southwest end of our profile, this could largely be due to inversion smoothing together with absence of station coverage. Finally, a thin conductive layer near the surface (upper 5 km) underlies all stations spanning the RIS. Its conductance (conductivity-thickness product) of ~1000 S is close to that of the estimated 300–400 m of seawater here[47], although some submarine sediments may be present.

## Discussion
The highly resistive lithosphere occupying nearly the top half of the upper mantle (TL) in Fig. 6 is strong evidence against active thermal support of the central TAM. The most resistive values in fact are closest to the rift shoulder front, reaching 10,000 ohm-m at ~80 km depth. In the active Great Basin of the U.S., the

Southern Sierra and the Wasatch Plateau that comprise the two margins exhibit upper-mantle resistivities of 100–200 ohm-m, which is consistent with peridotite mineralogy and an adiabatic upper-mantle geotherm[1, 4]. Our model tests overrule such a state as applicable to the central TAM. Moreover, basaltic magma ponding and releasing fluids near the Moho from upper-mantle convective melting tends to create strong conductive layers in the lower half of the crust[4, 39], which clearly do not appear below the central TAM in our study.

Consequently, we argue that the cantilevered flexural uplift hypothesis for the rise of the central TAM (Fig. 2c) is to be preferred. From previous discussion, this is largely by a process of elimination. The MT evidence is against a significant upward thermal load; as is apparent from the resistivity contrast to 150–200 km depth, little of the thermal regime of WA has encroached upon the central TAM. Simple isostatic uplift of the central TAM in response to regional WA rifting is not considered viable due to inconsistency in timing[14, 15, 36]. Crustal root contributions appear minimal here and for most of the TAM. This may warrant re-evaluation of the importance of cantilevered flexural uplift elsewhere along the TAM, even where thermal buoyancy loads can be entertained. A poorly constrained additional influence on central TAM buoyancy may be compositional density variation between EA and WA lithospheres.

The MT results for the central TAM contrast with recent seismic tomographic modelling, which depicts low S-wave velocity along a band ~200 km wide spanning the 60–120 km depth range in the uppermost mantle underlying almost the entire TAM[23]. Although we cannot speculate in detail on those model controls, seismic data sampling is sparse in our central TAM region (Fig. 1) and the seismic model is sensitive to the tradeoff between constraining TAM crustal thickness and uppermost mantle velocity, which was not specifically tested in our study area[23]. Hence, we support arguments that the long lateral wavelength of the central TAM, and much if not all of its uplift, result from cantilevered flexure against a steep regional border discontinuity or fault (BF in Fig. 6) separating EA from WA[40]. Flexure-induced movement on the WA side of BF may be

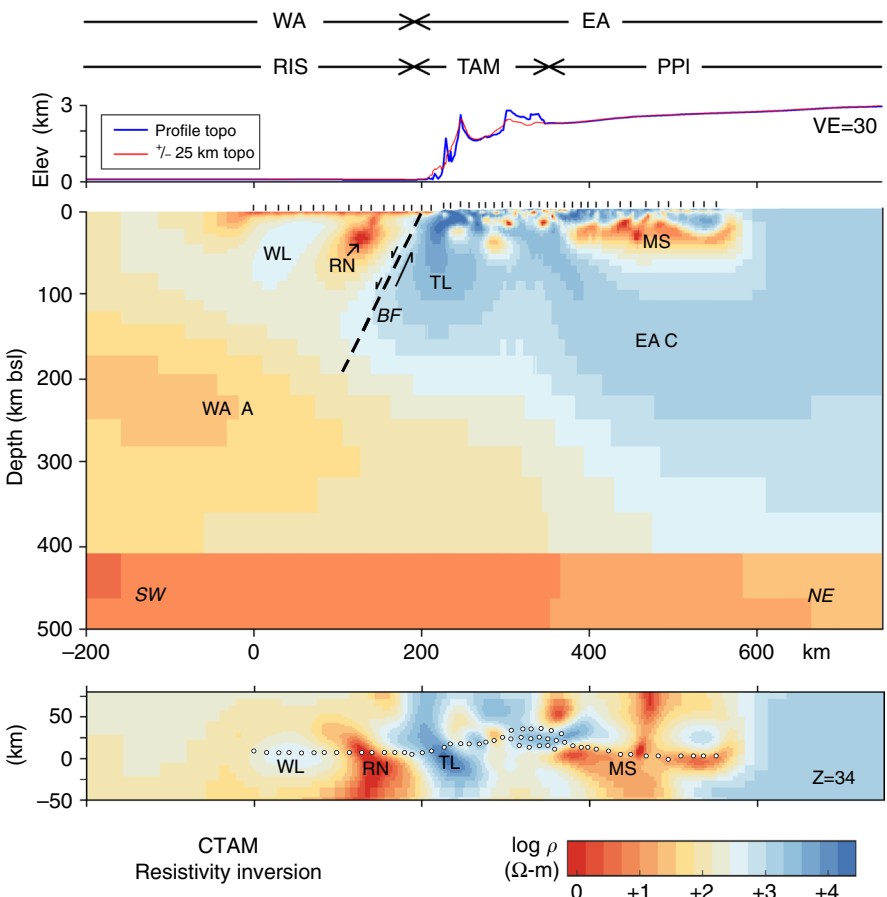

**Fig. 6** Three-dimensional resistivity inversion model for the central TAM. Physiographic regions are as in Fig. 2. Section view is slightly meandering to pass through stations of the main profile (Supplementary Fig. 11). Important model features interpreted include West Antarctic lithosphere (WL), active rift necking (RN), regional boundary fault (BF, schematic), TAM lithosphere (TL), Precambrian metasedimentary domain (MS), West Antarctic asthenosphere (WA A), and East Antarctic craton (EA C). Plan view in lower panel is shown at depth of 34 km over a width of 135 km, with MT stations as white dots. An additional plot of Fig. 6 using a spectral color scheme is presented as Supplementary Fig. 6

modulated by distributed rifting effects but no significant buildup of low-resistivity sediments appears. Slight subsidence may be consistent with excursion of the seawater-ice shelf grounding line northwestward beyond our study area[47]. Physical state elsewhere along the TAM of course may vary, particularly in the Ross Island —North Victoria Land segment.

The drastic change in crustal and upper-mantle resistivity moving southwest from the central TAM range front is interpreted to signify moderate late Cenozoic extensional activity in WA (Fig. 6). We argue that the compact, pronounced low-resistivity body RN approaching the TAM from the southwest in the nominally 15–60 km depth range reflects concentrated rift necking[38, 40]. If any portion of our model resistivity structure corresponds to basaltic melts underplating the crust and releasing fluids[4], it would be here. Lack of surface volcanism implies that the degree of extension below our transect is subdued compared to North Victoria Land and Ross Island regions[38]. However, feature RN (Fig. 6, plan view at Z = 34 km) expands toward Ross Island and North Victoria Land in the grid southeast direction where extension and volcanism are more pronounced. Farther southwest from structure RN, resistivities in the upper ~100 km are moderately high, nearing 300 ohm-m, signifying the WA lithosphere (WL). Such values together with lack of Moho-level, low resistivity from magmatic underplating and fluid release indicate subdued or dormant rifting farther from the central TAM.

At depths below 100–150 km under the RIS segment, broad-scale resistivity falling toward 30 ohm-m by 200 km depth (Fig. 6) is identified with the upper-mantle asthenosphere of extensional West Antarctica (WA A). This is not necessarily a zone of partial melt. Under the assumption of an adiabatic geotherm near 1450 ° C at 200 km, 30 ohm-m is consistent with the presence of hydrous defects having concentration on the order of ~200 ppm in otherwise solid-state peridotite mineralogy[39, 46, 48–51]. In this regard, the WA upper mantle resembles that of the U.S. Great Basin, where resistivity also falls at depths >100–150 km into a deeper zone of mantle hydration with a similar geotherm[39], although the WA extension today is much less vigorous. This deeper low resistivity is compatible also with a small degree of partial melt at somewhat higher temperature[51]. A 100 km thickness for the WA lithosphere (WL) compares reasonably to recent seismic S-wave estimates[23]. The absence of regionally extensive magmatic underplating with its latent heat release means that heat flow in this area of WA should be significantly lower than the U.S. Great Basin[4]. On the basis of its resistivity profile, heat flow appears on the order of 60–70 mW m$^{-2}$ similar to much of the Colorado Plateau interior, which has experienced little tectonism in the Cenozoic[4, 5]. This is compatible with seismic and Curie depth estimates for east-central WA, and is in keeping with the relatively low elevations of WA[23, 52, 53]. However, heat flow near the rift neck zone could be higher to an unknown degree. Cratonic heat flow of ~40 mW m$^{-2}$ pertinent to lithospheric

thickness of 150–200 km[54] should apply to the TAM and nearby polar plateau.

The low deep crustal resistivities toward the grid northeast end of our profile under cratonic Precambrian EA are interpreted to show the presence of large-scale metasedimentary bodies of possible Proterozoic age bearing graphite and perhaps sulfides (MS in Fig. 6). The high resistivities of cratonic lithosphere below (EA C) preclude a thermal cause. The primary source of graphite in such settings would be organic matter concentrated in sediment-starved forearc basins or continental margin rift basins transformed and possibly remobilized in metamorphism[55–57]. Where tied to outcrop, metasedimentary units in the central TAM are associated with low magnetization as revealed in aeromagnetic data, although visible graphite is not reported[58]. The most pronounced aeromagnetic lows project into the low-resistivity area MS, consistent with our metasedimentary interpretation. Metasedimentary units of the Paleoproterozoic Argosy Schists could be candidate lithologies, metamorphosed and carried to depth during the late Paleoproterozoic Nimrod or Cambrian Ross orogenies[10, 58]. Given the deep crustal locations of the conductors, these will not represent upper crustal, Phanerozoic sedimentary sections such as the Beacon Supergroup or other units of the Wilkes Subglacial Basin[59].

## Methods

**MT measurement technique on polar ice sheets**. Our soundings were acquired using commercially available (Phoenix Geophysics Ltd) V-5 and V-5a model MT systems. The magnetic fields were measured using high-moment induction coils similar to land operation. A "+" grounded bipole array was used for recording the electric fields. However, electric field fidelity is hindered by the very high contact impedance at the electrode-firn interface (0.5–2 Mohm commonly). This requires measurement modifications as we have outlined previously[43] with the following minor updates.

The distributed capacitance of a long bipole wire interacts with electrode contact impedance to create a voltage divider effect, $V_{in}/V = Z_{sh}/(Z_s + Z_{sh})$, where $V$ is the desired measurement and $V_{in}$ is the voltage presented at the first amplifier's input (normally the MT receiver input). Ideally, $Z_{sh} >> Z_s$ and $V_{in} \sim V$. Here $Z_s$ is a lumped series impedance that includes electrode contact impedance and $Z_{sh}$ is the shunt impedance at the amplifier input. $Z_{sh}$ can be modeled as the parallel combination, $(R_{sh} \| C_{sh})$, where $R_{sh}$ is the shunt resistance and $C_{sh}$ is the shunt capacitance[33]. A long bipole wire will increase $C_{sh}$ by many $n$ farad, introducing unwanted low-pass filter effects and allowing spurious noise input as contact impedance increases.

Comparable to our South Pole work[43], these issues were managed using custom buffer amplifiers (Numeric Resources LLC) placed at the electrodes to isolate the high contact impedance (Supplementary Figs. 1 and 2). For CTAM, we implemented simpler, non-inverting buffer amplifiers that integrated easily with commercial receivers. The shunt impedance at a buffer input is $R_{sh} = 90.9$ Mohm and $C_{sh} = 19$ pf; the output resistance is ~50 ohm. The single-ended output of a buffer connects to the appropriate receiver input terminal via the signal line of a shielded, twisted pair bipole. The bipole ground and shield drain wires connect to the receiver's ground electrode, with the shield remaining unconnected at the buffer. The buffer input was connected to an electrode using a short (~60 cm) wire "pigtail" (Supplementary Fig. 1), which increased $C_{sh}$ by ~10pf.

Preamplifiers were powered using 4 lithium-iron-disulfide AA batteries. For robust and electrochemically stable operation in the firn, we used 18 × 24 inch expanded titanium metal sheets buried horizontally. Bipole spans were ~150 m, laid out with regard for crevasse hazard (Supplementary Figs. 1 and 2). We deployed up to ten instruments simultaneously to take advantage of weather windows, with occupations of 4–12 days. Site transport was via helicopter, fixed-wing plane, or snowmobile, to sites commonly with rugged topography, which was accounted for in our inversion through the deformable finite elements (Supplementary Figs. 3 and 4).

The original field measurement coordinates were geomagnetic to reduce the chance of orientation errors during setup. For subsequent display and analysis, the $x$ axis was rotated to grid N315°, nearly parallel to the TAM. The required spectral rotations were small, ranging from −15° for the northeast profile end to −2° for the southwest end. Rotation angles were computed using the NOAA on-line calculator at http://www.ngdc.noaa.gov/geomag-web/#igrfgrid.

**Assessment of non-plane-wave effects**. Although the MT method depends upon the assumption of a planar, vertically propagating incident EM field, the convergence of solar wind-magnetospheric currents toward Earth's magnetic poles may lead to current concentrations in the oval polar electrojet proximal to our study area that challenge the plane-wave assumption[25, 60]. We argue from three standpoints that such potential problems have been avoided or removed.

First, the magnetic activity Ap indices[61, 62] for both field seasons show Quiet conditions prevailing (Supplementary Fig. 7). Only for 2 days in season one and 1 day in season two does the activity barely reach Unsettled. This is a positive sign for absence of strong current systems in the polar electrojet that could have affected our data.

Second, for several stations over the two field seasons, we examined spectral strength as a function of time during each recording. A representative example appears in Supplementary Fig. 8 for site B22 on the polar plateau. Although only computed to periods of several hundred seconds, a diurnal variability in ionospheric signal strength is apparent in the afternoon Greenwich time or early morning local time, especially in the horizontal magnetic fields $H_x$ and $H_y$. The likeliest candidate for this behavior appears to be the polar electrojet solar quiet variation[63] ($S_qP$), where higher intensity intervals represent closer approach of the auroral oval limb with concentrated, field-aligned, ionospheric current flow. Steadiness in such behavior is characteristic of a lack of solar magnetic disturbances.

Third, to test whether this signal behavior could represent diurnal variation in non-plane-wave contributions, we subdivided the total recording into alternating 12 h segments containing the highest and lowest activity segments (Supplementary Fig. 8). The high and low activity segments were grouped and underwent cross-site remote reference response processing using robust outlier removal with instrument vendor-supplied software[64] to create two independent MT soundings representing the two activity levels. These soundings are presented in Fig. 3.

Apart from minor scatter, the prior soundings are equivalent in major character over their entire period range. Similar results were obtained for other test soundings on the Ross Ice Shelf (RIS) and within the TAM for both seasons. From this we conclude that non-plane-wave effects are of negligible influence in our MT responses and we processed the entire time series to obtain soundings for use in inversion. For production processing, soundings were typically cross-remote referenced for random noise abatement although a dedicated reference near the CTAM camp was run during season one.

**Impedance phase tensor and induction vector representations**. Resistivity structural trends and dimensionality are commonly illustrated using phase tensors derived from the MT impedance, and induction vectors derived from the vertical magnetic field or tipper[25, 65, 66]. We have performed this for five periods spaced roughly one decade apart in Supplementary Fig. 9a and b. If the study area is approximately two-dimensional, one should see either the major or minor axes of the phase ellipses be aligned NW-SE parallel to the TAM range front. One also should see the induction arrows aligned roughly perpendicular to the range front for 2D conditions.

Because phase tensor ellipses contain no impedance amplitude information, the fundamental differences between the RIS and the TAM or polar plateau (PPl) are not represented in the manner of Fig. 4. However, the ellipses over the RIS still show striking differences in character compared to within the TAM or PPl (Supplementary Fig. 9a). With rare exception, the RIS phase tensor ellipticities and skews ($\beta$)[65] are low suggesting that the departure from 1D conditions for these sites may not be large. Given overall levels of $\rho_a$, relatively high values of the phase tensor determinant $\Phi_2$ at shorter periods should denote sensitivity to lower resistivity below the RIS ice sheet[65]. Relatively low values of $\Phi_2$ at middle periods denote sensitivity to higher resistivity in the deep crust. The $\Phi_2$ values rise again at the longest periods indicative of lower resistivity again, which we show through non-linear inversion to exist in the upper mantle below the RIS. However, at the longest periods the axes of the RIS ellipses become oblique to the profile orientation (Supplementary Fig. 9a).

Moving onto the TAM and PPl, phase tensor ellipses change drastically. Although there are intervals along the profile where ellipses group somewhat in their alignment, $\Phi_2$ values and ellipticity, there is commonly great diversity in these parameters especially over the cluster of sites near the TAM-PPl transition (Supplementary Fig. 9a). Some ellipses are virtually linear. This behavior is not due to statistical data scatter and can persist to the longest periods, corresponding commonly to out-of-quadrant phase excursions first noted with the pseudosections of Fig. 4d, m and Supplementary Fig. 5d, m. Correspondingly, there are many instances where the phase tensor skew ($\beta$) is not only large in magnitude but becomes negative[65, 66]. In this region, few sites have $\beta$ lying within a recommended 5° criterion for a possible 2D situation[66].

The induction vectors (Supplementary Fig. 9b) are strongly affected by the TAM transition given that a component of them point to a conductive zone just grid southwest of the range front. However, the vectors in this area are oriented even more strikingly toward grid southeast indicating that a strong gradient in conductivity exists with values increasing that way. This is represented with the strong positive anomaly in $Re(K_{zx})$ seen in the pseudosections of Fig. 4p and Supplementary Fig. 5p. A large low-resistivity axis near the grid northeast end of the transect in the PPl in presumably cratonic rocks is observed as well. Numerous local conductive anomalies at diverse locations beneath and lateral to the profile also are in evidence. The numerous 3D effects in the observations warrant rigorous analysis in creating a model of subsurface resistivity.

**Upper crustal and ice structure constraints**. Away from the central TAM front, MT soundings on the Ross Ice Shelf exhibit nearly isotropic behavior (Fig. 4) suggesting that 1D analysis may yield useful results at least at shallow levels and provide constraints for 3D inversion procedure. To this end we display the response from site A21 at the southwest end (Supplementary Fig. 10) and perform layered and smooth 1D inversion. The finite layered algorithm uses a Marquardt approach[67] written in-house and applied previously to other settings[4]. The smooth inversion uses the same forward and Jacobian computations, but invokes many layers linked together for spatial smoothing comparable to the 3D. This approach is meant to provide maximal flexibility in fitting the observations. Because the results are nearly isotropic, we invert the impedance invariant $|Z_{xy} − Z_{yx}|/2$ in $\rho_a$ and $\varphi$ form. Errors floors of 0.03 log units $\rho_a$ and 2° in $\varphi$ are imposed as representative of scatter and to help equalize fit across the spectrum.

The finite layered model has five primary units (Supplementary Fig. 11) intended to represent bulk properties of the ice, seawater, any subsea sediments, deeper crust, and upper mantle. The resistivity of the seawater was fixed at 0.3 ohm-m[68] although its thickness was allowed to vary. The starting value for the other layers was 50 ohm-m. The computed response of the final model plotted in Supplementary Fig. 10 shows good fit to the data with normalized root-mean-square error (nRMS) of 0.92, where unity is ideal. The bulk resistivity of the ice is only just over 400 ohm-m, vastly lower than the polar plateau values observed earlier near South Pole[43]. In addition to comparatively higher temperatures at the RIS, we suggest this may be attributed to proximity of the Ross Ice Shelf to the Southern Ocean and the introduction of salt-related impurities to the ice boundaries. Nevertheless, the contact impedance at the titanium electrode interface remained high, of order 250 k-ohm presumably reflecting the porous near-surface firn. The thickness near 400 m is compatible with earlier work estimating ice shelf thickness using radar and seismic refraction[47].

Below the ice layer, the nominal seawater thickness is ~150 m (Supplementary Fig. 11), which is less than the 300–400 m range of seawater estimated in earlier surveying[38]. Below the seawater model layer is another layer somewhat over 2 ohm-m and ~2400 m thickness. Tentatively, this could be assigned to a layer of marine sediments. On the other hand, its conductivity-thickness product (conductance) is similar to that of the seawater layer and the two layers together sum in conductance to approximately that of a 300 m thick seawater layer. A separate four-layer inversion was run to see if a thicker seawater layer without sediments below could explain the data equally. The resulting nRMS was 1.90, which significantly, though not greatly, exceeds that of the five-layer model. At face value, that is a less preferable model but is tempered somewhat by the moderate conflict of the five-layer model with the radar/seismic data. It may be that we are near a transition in seawater thickness[47].

The smooth model inversion starting from a 50 ohm-m half-space achieved an nRMS of 1.27 and its results are plotted in Supplementary Figs. 10 and 11 as well. Its character is compatible with the layered results including ice resistivities of a few hundred ohm-m, moderately resistive crust of several tens of ohm-m, and a drop in resistivity in the shallow upper mantle. A local resistivity low just over 2 km deep presumably represents possible submarine sediments, and this feature only becomes more extreme as regularization is relaxed. As a result of the overall 1D examination on the Ross Ice Shelf, our subsequent 3D inversion incorporates a fixed ice layer of ~300 m thickness and 300 ohm-m resistivity. However, we do not impose a hard constraint on a seawater layer but allow a smooth (regularized) conductive layer to form in the 3D model without specifying whether it all is seawater or a combination of seawater and sediments.

**Construction of finite element inversion model**. The total finite element mesh, whose central portion is depicted in Fig. 5, is of size 61 × 257 × 67 (x–y–z) elements with the x-direction oriented grid N315°, parallel to average trend of the central TAM. Of these, there were 14 layers assigned to air. Each subsurface finite element is a cell in the inversion model, except for a rim of fixed elements around the mesh margins. Horizontal cell dimensions along and near the receiver array are in the 2 × 2 to 3 × 3 km range, and the smallest vertical cell dimension (near-surface) is 260 m. Outside the data coverage, the mesh horizontal sizes expand gradually and geometrically to mesh edges ~1800 km from array center. Topography data for the mesh vertical deformation are from the 400 m RAMP resource[69], which were averaged laterally to provide values for the element corners. Our finite element formulation assumes that the model asymptotes to a 1D background at the mesh edges so an elevation of 1000 m was chosen as an average to bound the model domain.

All 57 stations were included with 33 frequencies per site and 12 data (four complex impedance and two complex tipper elements) per frequency for a total of 22,572 data. Data error floors imposed were 0.05 for both real and imaginary parts of $K_{zx}$ and $K_{zy}$, and 5%max{$|Z_{ij}|;|Z_{ival}|$} for ($Z_{ij}$), where $Z_{iva} = (Z_{xy} − Z_{yx})/2$ is an impedance rotational invariant[10]. The starting model was a 300 ohm-m half-space (similar to average over all $Z_{iva}$) down to 410 km, beyond which 20 ohm-m was the start based on laboratory constraints across this mantle phase transition[46]. All inversion cells were allowed to vary with exception of an ice layer over the Ross Ice Shelf fixed at 300 ohm-m for the upper cell layer (260 m) and 100 ohm-m for the next layer down to approximate the previous 1D analysis results. Inversion models were computed on a 24-core Linux workstation with 0.5 TB RAM as described for the algorithm development[44, 45].

The initial nRMS with the inversion starting model was 50.5. Inversion was allowed to progress until nRMS reduction per iteration became <10% whereupon the regularization parameter $\lambda$[45] was halved. We computed inversion models with and without estimates of the impedance static distortion at each MT sounding[45]. Distortion parameters were estimated in the latter portion of the model convergence. The best fit without distortion estimation occurred at model 16 with nRMS of 1.67. Subsequently, distortion parameters were estimated starting with model 13 and a final nRMS of 1.43 was achieved at model 17. This is the model displayed in the main text in Fig. 6. The relatively minor improvement in overall misfit with static distortion estimation being consequential at only a few sites, principally in the site cluster near the TAM-PPl transition (Supplementary Fig. 12). Otherwise, discretization of the mesh was sufficient to represent most small-scale response distortion local to the MT stations. One exception is the site near center of the TAM-PPl cluster where final nRMS exceeds 3. The fit is good at the shortest periods, but by 1 s a constant offset in observed vs. predicted $Z_{xx}$ and $Z_{xy}$ develops. This represents structure apparently too small to be sampled by the voxels of our model in this area, but too large to behave in a purely static fashion.

The model without static shift estimation is displayed in section view in Supplementary Fig. 13. The models also are compared in a series of plan views in Supplementary Figs. 14 and 15. One sees only minor differences with the preferred model in Fig. 6 inconsequential to the tectonic interpretation, in keeping with static distortion affecting just a few stations. Much of the difference in fact could be the result of the two inversions ending on slightly different model numbers. We do note that using a smoothing regularization in the inversion causes both breadth and depth extent of structure to become increasingly smeared offline of the profile. In particular, the faint drop in resistivity in the model section views ~100 km below the TAM-PPl transition (Fig. 6; Supplementary Fig. 6) represents influence of offline structure northward at crustal depth at those lateral distances and is not a deep thermal effect. Additional offline MT sounding data would be required to improve resolution there.

**Two-dimensional resistivity inversion model**. Given that the data coverage primarily is in profile form, 2D inversion was attempted for comparison with the 3D using a separate in-house algorithm with similar regularization[4, 70]. Because one usually has poor control on the strike extent of structures beneath a field profile especially with 2D data, for 2D inversion it is recommended to emphasize the nominal transverse magnetic mode of impedance ($\rho_{yx}$ and $\varphi_{yx}$) (electric current flow along the profile) plus possibly the tipper element $K_{zy}$. These were selected from the main profile of the CTAM dataset with $\rho_{yx}$ and $\varphi_{yx}$ given the same error floors as in the previous 1D inversion plus a floor of 0.05 for the real and imaginary components of $K_{zy}$. Impedance values were omitted if $\varphi_{yx}$ reached or exceeded 90° as such are not permitted in a purely 2D TM mode[71]. No attempt was made to fit $\rho_{xy}$ and $\varphi_{xy}$. Because it is much less demanding of computer resources, the shallowest mesh elements in the 2D model were only 40 m and data to 100 Hz were utilized. The starting nRMS was 44.1, ending at 4.02 in eight iterations. Quantitatively the final nRMS is much greater than that achieved in the 3D inversion, attesting to the rigor of the 3D calculation, although visually the 2D fit appears reasonable (Fig. 4c, f, i).

The 2D model in Supplementary Fig. 16 shows similarity to and differences from the 3D section of Fig. 6. Resistive lithosphere under the TAM range front area persists still to at least 100 km depth. One observes a lower resistivity region under the RIS at depths beyond 100 km that would represent the WA asthenosphere. It is more weakly represented than in the 3D case we believe because the lateral resolving contribution from the $xy$ component of the impedance is absent in the 2D. A low-resistivity rift neck structure appears again just southwest of the TAM front. A strong tabular conductor in the middle crust occurs under the northeast end of the profile representing the major metasedimentary package interpreted there. High resistivity below the northeast limit of the MT transect corresponds again with EA cratonic lithosphere. Resistivity of the deeper lithosphere under the middle TAM area is more obscure, with intermediate values taken to reflect a smearing or "sideswipe" of major conductors offline that are resolved appropriately in the 3D inversion. Although providing a useful image, the poorer model fit, reduced data incorporation, and shortcomings in dimensionality assumption make the 2D model inferior to the 3D.

**Tests of resistivity model resolution**. Our initial test of MT inversion resolution is to illustrate depth of penetration of the MT signals. To do this, we replace the 20 ohm-m material below 410 km in the original starting model with 300 ohm-m like the starting material at shallower levels, and keep this deeper material fixed in a re-inversion (not using static distortion estimation for simplicity). The resulting model appears in Supplementary Fig. 17 and generated a final nRMS of 1.667, negligibly different from that of Supplementary Fig. 13. However, one sees that this latter model contains regions of much lower resistivity in the deepest upper mantle adjacent to the 410 km discontinuity, especially under the TAM and PPl. These have been emplaced presumably to mimic the effect of the original low-resistivity (20 ohm-m) material below 410 km in the starting model. Given that lower resistivity below this phase transition is considered well-founded[46], we conclude that our signals penetrate at least to this depth because omitting this conductor required an ersatz conductor to form in the model above.

A key conclusion of our study is that highly resistive, craton-like properties exist under the TAM right to its range front and that elevated thermal conditions in the shallow mantle and lower crust do not exist there. To test that this is preferred to the alternative, we erase resistivities greater than a conservative 300 ohm-m below the TAM to the TAM-PPl transition (Supplementary Fig. 18). These values are toward the upper end expected for dry peridotite under average mantle adiabat temperatures (more like 100–200 ohm-m[46, 48]) so that a thermally disturbed lithosphere should, if anything, be even less resistive. The degradation in fit of this alternate model to the data is represented in Supplementary Fig. 19 on a site-to-site basis. Because these deeper structures affect longer periods, we specifically examine quality of fit for $T > 1$ s. One observes a significant increase in misfit especially in the vicinity of the TAM front. From this, we reject thermally elevated resistivity properties as being acceptable beneath the central TAM region.

A second important structure in our interpretation is the low-resistivity rift neck feature (RN), which would represent a concentration of extension at the margin between two lithospheres (WA and EA) of differing strength. We tested its importance in the MT response by replacing it with 300 ohm-m material similar to surrounding lithosphere (Supplementary Fig. 20). Again, the degradation in fit of this alternate model to the data as represented in Supplementary Fig. 19 on a site-to-site basis is unacceptably high for entertaining uniform deep crustal and uppermost mantle properties through this area. Thus, the RN feature appears to be a requirement of the data.

**Data availability**. The MT responses are archived with NASA Global Change Master Directory at http://gcmd.nasa.gov/getdif.htm?NSF-ANT08-38914 in accordance with USAP policy.

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

## Acknowledgements

The Antarctic fieldwork and data analysis of this project were supported by U.S. National Science Foundation Grants OPP-0838914 and OPP-1243559 to P.W. G.H. was supported under the GNS Science AMT program and under Royal Society of New Zealand Marsden Fund award ASL 1301. We are grateful to the U.S. Antarctic program for logistical support within the multi-investigator CTAM field camp of season one and the dedicated RISC fly camp of season two. Field safety/mountaineering support was provided by J. Pierce, S. Norman, M. Smith and J. Hanna, and by D.U. Any use of trade, product, or firm names is for descriptive purposes only.

## Author contributions

The experiment was designed by P.W. Field logistics were lead by P.W., G.H. and Y.O. Home institutions of P.W., G.H., Y.O., T.B. and G.B. provided instrumentation. J.S. designed and built the high-impedance pre-amplifiers. Time series processing was carried out by G.H. V.M. computed the inversion models and most graphics. P.W., G.H., J.S., V.M., Y.O., K.S., G.B. T.B., D.U., B.A., A.M.G. and D.F. (all authors) participated in the field campaign and provided input to the manuscript.

## Additional information

**Competing interests:** The authors declare no competing financial interests.

