## [Peer Review File · Nature Communications]

Reviewers' comments:

Reviewer #1 (Remarks to the Author):

This manuscript presents new results of a magnetotelluric (MT) survey conducted in a profile across the central Transantarctic Mountains of Antarctica in order to evaluate models of mountain range uplift. This is a long-standing enigma that has been addressed by field geologic studies, seismic surveys, gravity modeling, low-T thermochronology, and numerical modeling. Despite a great deal of attention, no satisfactory explanation has emerged and there is no consensus.

This paper brings a fresh set of observations to the site of arguably one of Earth's most prominent surface and geotectonic features. The authors set out to explain this enigmatic feature of the Antarctic plate, and they set up the problem as a test of 3 competing hypotheses regarding uplift mechanisms. The conclusions of the paper — that of flexural uplift during regional extension — is valid and supported by the observations. The application of a unique geophysical approach (novel for this problem) is valuable, especially due to the remoteness and extensive ice cover in Antarctica, which makes other methods less tractable.

In general, I think this is a very strong contribution and provides compelling results. Authors do a good job explaining the range of features apparent in their data, and they conclude with a valid argument for flexural uplift based on inferred mechanical properties. Some questions about interpretation are listed below. I also provide a marked-up manuscript with suggestions for improving the clarity and succinctness of the writing. I recommend publication pending appropriate revision.

Writing: Lots of suggestions for revision. Word choice matters — a number of substitutions are recommended, in some cases to correct improper usage, and in some cases to clarify or prevent misinterpretation.

Comments by line number:

1. I suggest deleting the first two words "Origin of". The title has two 'ofs' which is a bit awkward, plus you don't want to convey that this is about origin of the TAM themselves (a different problem). A cursory glance might leave the reader confused. Why not just keep it simple? — "Uplift of the Central TAM"?

25-27. See comments about order of ideas in the summary. Perhaps it is recommended for this journal, but the sequence of points is not logical to me.

38. See comment about use of the term "compositional". A bit ambiguous as used.

58-59. Unsupported statement. References suggested.

92. Not clear what this sentence is driving at — obtuse. Does this mean that the seismic data 'minimize importance of root buoyancy', which in turn eliminates this idea as a contender? Or is the MT data that do? Latter doesn't make sense because it says above that MT data are of limited help. Be explicit. Perhaps authors mean something like: "Nevertheless, seismic data indicate that root buoyancy plays only a minimal role in uplift of the central TAM and leads us to consider other geodynamic mechanisms."

107-110. This sentence is very confusing. From the language used, it's not clear if EA currently has an elastic thickness of 85 km, or if this has actually been reduced by some mechanism. Use of "would be" is central to this confusion. Is it, or isn't it? If it is reduced, how does this relate to the flexural uplift model? Further, it is confusing what is meant by "from WA" -- again, is it being argued that EA was influenced by heat transfer from WA? Is this what flexural modeling predicts,

or is this a hypothetical? This whole concept needs clarification.

212. See comment about use of geographic directions. Antarctica is unique and requires care to avoid ambiguity.

224-230. Like remarked for WA (lines 224-230), can the authors predict heat flow for EA lithosphere inboard of the TAM? This would be very useful for interpreting geophysical data, geological interpretation, glaciology, and ice-sheet stability.

231-241. Very interesting comparison of the MT data with outcrop and aeromagnetic studies. It makes general sense that the low resistivity materials are relatively high-grade metasedimentary rocks like those seen in the Nimrod Group with abundant hydrous minerals but not graphite (however, note, the metasedimentary units have recently been distinguished from gneissic rocks of the re-named Nimrod Complex (not Group); metased rocks are now referred to as the Argosy Schists; Goodge and Fanning, 2016, Precambrian Research). Beyond the terminology, just for sake of argument, can authors preclude that the material with low resistivity inboard of the TAM is not made up of relatively young sedimentary basin (e.g., Wilkes basin with Phanerozoic sediment, not Precambrian)?

General comment about 3D interpretation (Fig. 4) — I think it would be important for the authors to comment on the interpretation of the bounding rift fault shown in Fig. 4 as a steeply W-dipping structure, whereas the MT data in Fig. 3 show essentially vertical gradients between W and E domains as depicted in the depth profiles. Why does the interpretation show a W-dipping lithosphere scale structure when the data do not? This could be simply a matter of depth resolution of the MT data and how they propagate, that the signal is attenuated, or some other justifiable cause. However, I note that in the horizontal depth slice at 34 km in Fig. 4 (lower panel), the resistivity 'anomalies' tend to track the MT stations, where many of the anomalies share the same orientation as the profile and the sharpest gradients are transverse to the profile. Because there is not a 2D grid of surface stations, this suggests that at least some of the anomaly pattern is due to uncontrolled interpolation of results in the direction away from the profile where data are lacking (a far-field effect). If, then, the profile line acts as an attractor of anomaly gradients in X-Y, then how well constrained is the vertical structure in Z?

Figure 1. INSET: Change color of SP and RI to white. Label WA and EA. MAIN PANEL: Label main glaciers in the main panel — Beardmore and Nimrod. Label Ross Ice Shelf. Change CTAM and RISC either to colors that match the transect stations, or make them the same (random colors are not helpful). Caption should indicate that two geographical grids are used.

Figure 2.

(a) Overall, a well constructed figure depicting alternative hypotheses. The essential elements are clear in panels a and b; in c, it might help to make a stronger color contrast between mantle lithosphere in the WA and EA portions, which will help emphasize that lithosphere in the two blocks is of contrasting character.

(b) On line 108, the text refers to an elastic thickness of 85 km — see note above, but in any case if the elastic thickness (strong layer) is less than the 100 km depth shown here, perhaps add a dashed line to all diagrams showing best estimate of depth to viscous mantle?

John Goodge

Reviewer #2 (Remarks to the Author):

Summary

This manuscript describes the results of a magnetotelluric (MT) survey across the central

Transantarctic Mountains (TAM), aimed at investigating the uplift mechanism for this extensional mountain range. Results indicate high resistivities to ~150 km depth beneath the TAM, which are interpreted as cold, stable material extending into the upper mantle. The authors interpret this structure to be most consistent with a flexural cantilever model, in which thermal influence plays little role in supporting the TAM.

Before outlining my questions and comments on the manuscript, I want to emphasize that I am not very familiar with MT analyses. Therefore, I cannot provide a critical assessment of this manuscript in terms of the methodology (most of which is highlighted in the supplemental information). Given this, most of my review is focused on the concepts and interpretations presented by the authors.

While I found this manuscript to be very interesting, I have a number of comments/concerns that make me question if Nature Communications is an appropriate venue for this publication. Specific comments are outlined below, but my main criticisms can be summarized with a few key points. These include:

a) A variety of previous work has been done in the TAM, particularly in regard to different uplift mechanisms, and numerous models have been proposed. The discussion of these different models, and more importantly references to the studies that developed these ideas, are too limited as presented in this paper. Even the flexural cantilever model, which the authors advocate for, is not well described. This uplift mechanism strongly depends on the elastic lithospheric thickness, potential heat sources, and ice loading and more development could be done to integrate the MT findings with this model.

I realize some of the above may be due to page and reference number limitations imposed by the journal, but I think the work suffers from this limitation. A more complete discussion of these uplift models, the studies and types of data used to suggest them, and other components that could contribute to TAM uplift would improve the manuscript.

b) Both the abstract and later discussion in the manuscript suggest that the flexural cantilever model, where thermal influence from the upper mantle is insignificant, is the best and most viable model for the TAM as a whole. However, inconsistencies between the results from this study and those from previous investigations (outlined below) would suggest that this likely isn't the case. Along-strike variability certainly has a substantial impact on the TAM uplift. That being said, even within the study area itself, certain characteristics of the TAM subsurface structure are too readily dismissed or over-simplified. For example, the current dataset provides no constraints on crustal thickness, but the authors dismiss any influence this may have on TAM uplift. The examined models also only consider the case where anomalous upper mantle material from West Antarctica physically extends beneath the TAM front, but thermal conduction from hot, neighboring material is not addressed.

c) Related to the point above, there is significant disagreement between the results of this study and previous investigations, particularly seismic studies. The authors provide little discussion on why these discrepancies may exist. Since this manuscript is advocating for a significantly different interpretation, the brunt of the responsibility to address these alternative viewpoints is on the authors. The manuscript is lacking in this aspect.

d) More generally, the discussion and figures in the main text rely too heavily on the supplemental information (SI). That is, many of the parameters included in the discussion and shown in Figure 3, in particular, are not well defined. Unless the reader is well-versed with MT analysis, many of these details are confusing. For a broader-readership journal like Nature Communications, the paper (main text and figures) should be able to stand on its own, with the SI providing additional details for those interested in the specifics.

The following outlines specific comments on the manuscript, with line number references provided. Some are small wording suggestions, which are less critical, but others highlight issues in relation to the points noted above. All of this being said, my recommendation is that this manuscript would likely be better suited for a longer-formatted journal, where the authors can more completely address the issues outlined in this review.

Specific Comments

Line 33: Suggest replacing "held up" with "supported"

Lines 39-43: It is somewhat disingenuous to say "the universal applicability of a thermal cause for rift uplift". A wide variety of mechanisms have been proposed to uplift and support extensional mountain belts. For the TAM, this is just one proposed uplift mechanism (from Stern and ten Brink, 1989 – which the authors fail to reference). Other studies, such as Fitzgerald et al., 1986; Chery et al., 1992; Studinger et al., 2004; Lawrence et al., 2006, have all put forth alternative models. As the manuscript is currently written, I think it does not give enough credit to these other studies and the mechanisms they have suggested. That is, the authors are not the first to suggest the TAM are supported by a non-thermal mechanism.

A bit further into that sentence, the authors also state that "...the presence of thermal activity potentially could obscure other causes..." While a valid point, the contribution of other characteristics leading to uplift does not necessarily preclude a thermal component from playing a significant role. Additionally, the authors themselves do not really consider other contributions; they simply try to rule out the thermal aspect.

Line 49: "WA" and "EA" are never previously defined as West Antarctica and East Antarctica. They should be.

Line 59: The text "WA was assembled to EA..." does not really make sense. The wording makes it sound like WA turned into EA. Clarify.

Lines 64-65: It would be good to more explicitly state when the TAM uplift began (~55 Ma) and reference some of the apatite fission track analysis that constrain this estimate.

Line 70: The seismic models being referenced and discussed are not "regional models." They are continental-scale surface wave studies.

Line 74: The word "the" should be inserted before "Ellsworth-Whitmore Mountains". Similarly, "the" should be inserted before "Ross Island".

Line 75-78: The comment that "most of the TAM rangefront [is] still coarsely sampled" is not specifically true. Admittedly, from the central TAM toward South Pole, seismic coverage is more limited. However, the POLENET and TAMSEIS deployments provide reasonable coverage of the central TAM, and a newer deployment (TAMNNET) provides better coverage of the northern TAM, near Terra Nova Bay. Together, constraints from regional studies based on these data have shown persistent, slow upper mantle velocities along the TAM front from Ross Island up to Victoria Land (e.g., Watson et al., 2006; Lawrence et al., 2006; Heeszel et al., 2016; Graw et al., 2016; Brenn et al., 2017). That is, the slow upper mantle velocities are not just concentrated beneath Ross Island, so the description here is misleading.

Line 92: The word "the" needs to be inserted before "importance".

Lines 80-81: The authors note that uplift models for the TAM generally fall into three categories, though no references for these models are provided in the text nor in corresponding Figure 2. As noted previously, more credit needs to be given to prior work in this regard.

Lines 84-93: The first of the three uplift mechanisms described by the authors (and illustrated in Figure 2a), which would require a crustal root beneath the TAM to provide isostatic compensation for the mountain range, is ruled out. Comparing this to the text on Line 67, there is a bit of a contradiction here. In that earlier section, the authors reference Huerta and Harry (2007) to make a case for lithospheric rift necking along the TAM, but the Huerta and Harry (2007) model argues for a thick crustal root beneath the TAM, which is now dismissed in this later text.

Further, this mechanism is ruled out not based on the authors' work or data (which they admit) but rather based on other geophysical studies. For instance, receiver functions (Hansen et al., 2009) suggest little to no crustal root beneath the TAM, but these "point measurements" correspond to stations further grid south than the study area examined in this manuscript. The authors also reference the gravity study of Block et al. (2009) to infer crustal structure and suggest that this study does not advocate for thick crust beneath the TAM either. However, the Block et al. (2009) study argues for "thick crust along the full length of the Transantarctic Mountains" with "crust 4-9 km thicker than the hinterland." That study argues that thick crust invokes an isostatic mechanism. The upshot: I don't know if the authors can justifiably rule-out this mechanism because (a) they cannot constrain crustal thickness with their data (which they admit) and (b) the only other local measurement of crustal thickness in their study area (i.e., gravity constraints from Block et al., 2009) actually does advocate for thicker crust. It may very well be that little to no crustal root is present beneath the central TAM, but this study cannot reliably dismiss this option for the study area being considered.

Line 97: The Stern and ten Brink (1989) reference would likely be more appropriate here, as opposed to the ten Brink et al. (1997) reference, as the other paper was the first to suggest this uplift mechanism.

Lines 99-101: The authors note that slow upper mantle tomographic structures have been proposed in support of a thermal buoyancy mechanism that contributes to the TAM uplift, but then they also state that "most of these studies focused on the north-central Victoria Land/Ross Island region where upper mantle is anomalously slow." This could be argued for regional studies, such as Watson et al. (2006), Graw et al. (2016), and Brenn et al. (2017) – which are not referenced in the manuscript – but the authors instead reference Hansen et al. (2014) and Heeszel et al. (2016), which are continental-scale body wave and broad regional-scale surface wave models, respectively, that do provide coverage of the study region.

Resolution of the Hansen et al. (2014) model in the study area is reduced compared to some other regions due to fewer seismic stations and the nature of body wave tomography. That is, the shallow velocity structure is more difficult to resolve with this approach, so a comparison to this study may not be the most applicable. However, the surface wave approach of Heeszel et al. (2016) allows for high resolution of shallower seismic structure, and this model does show anomalously slow upper mantle beneath the study area. Indeed, they report "a region of slow seismic velocity in the upper mantle (60-120 km depth) follows the TAMs from the Ross Island region through the central TAMs." From the seismic constraints, there is good evidence that the slow upper mantle is not just concentrated beneath north-central Victoria Land and the Ross Island region. This could also be interpreted to play an important role in the uplift of the central TAM. More on this in later comments.

Lines 101-104: The comments about MT and the comparison to the U.S. Great Basin seem out-of-place here. Consider moving this text elsewhere or rewording to make the transition between thoughts more clear.

Line 105-110: Several comments on this text:

a) Uplift along the TAM is variable. For instance, Stern and ten Brink (1989) estimated 5 km of uplift while ten Brink et al. (1997) instead estimated 7-8 km of uplift. Even larger uplift estimates

have been put forth by other studies. Is the 5 km of uplift noted by the authors specific to their study region? If so, what is the reference for this?

b) (Line 108-109) The effective elastic thickness (T_e) of EA from ten Brink et al. (1997) is not an observation. Instead, this characteristic was computed/inferred from flexural modeling carried out by that study as it was required to fit both the TAM topography and gravity measurements. That model and its corresponding calculations include a thermal buoyancy component (i.e., mechanism #2). In other words, the reported 85 km T_e estimate on Line 109 already includes a "lateral warming" source from WA, so I don't think the conclusion that it would be "reduced substantially" (line 108) is valid here.

c) It is also worth pointing out that the T_e estimate on Line 109 is based on a total uplift for the TAM of 7-8 km (ten Brink et al., 1997), not 5 km as the authors note on Line 105. The T_e estimate greatly depends on the total uplift being modeled. For instance, in an earlier study by Stern and ten Brink (1989), a similar flexural model was used to evaluate total TAM uplift on the order of 5 km, and in that case, the computed T_e was 115 +/- 10 km. Similar to ten Brink et al. (1997), that T_e estimate is also determined by including a thermal component in the uplift model.

d) Given the limited discussion of the flexural cantilever model, the authors do not describe how these model constraints directly relate to their MT observations (or vice versa). That is, the connection between the MT work and the advocated model is lacking (note: this applies to later discussion as well).

Line 119-120 (and associated text in supplemental): As already noted, I'm not very familiar with MT processing. Given the high impedance of the firn layer, it sounds like prior work at South Pole employed custom buffer amplifiers, but the authors state that the current work used "simpler, non-inverting buffer amplifiers." Why is the approach for this dataset different than that used in their prior polar work? Another question: are there MT constraints elsewhere along the TAM front?

Line 126: The authors say they resolve resistivity through the upper mantle. To what depth, specifically? Are certain depths resolved better than others? More specifics on these details are important to the interpretation.

Line 127-132 (and associated text in supplemental): Are there considerations that come into play when transforming off-diagonal impedance values to apparent resistivities? For instance, on Lines 131-132, it says that this provides a "smoothed version of subsurface resistivity structure." How smoothed? What is the corresponding resolution? Does this depend on how the apparent resistivities are computed?

Lines 154-155: The text states that sensitivity tests indicate the longest period data are sensitive to the 410 km phase transition and thus define the "minimum depth of survey penetration." However, looking at Figure 4, it appears that the model is fixed at this (410 km) depth. It seems suspect that such a perfect boundary would be resolved by the data and exactly at 410 km depth. So I have concerns over how the model is parameterized and what is really being resolved in the mantle.

Lines 158-160: The authors note that they test the high resistivity TL feature in Figure 4 by imposing lower resistivity values in this region and that the misfit correspondingly increased. This low resistivity material was extended ~150 km beneath the TAM front, but most seismic investigations would suggest that if a thermal anomaly is present under the TAM, it extends 50-100 km inland, at most. How would the results change if a structure more comparable to this were tested? Taking this a step further, how would the authors represent thermal conduction? That is, what if anomalous upper mantle material (i.e., a thermal anomaly) didn't actually extend beneath the TAM but rather provided a source of heat that affected the neighboring region? All of these options have been suggested in earlier investigations and models.

Lines 173-174: Given the comment here about inversion smoothing and the absence of station coverage, is it reasonable to show results off-profile to the SW and NE? Perhaps the lateral extent

of Figure 4 should be trimmed to focus on the area beneath the MT survey.

Line 177: "nRMS" is never defined. It should be.

Lines 179-180 and 189-191: As already noted above, this conclusion (i.e., against thermal support) does not readily agree with seismic constraints in this region. The authors themselves point this out on Lines 189-191, though for reasons outlined previously, I think a reference to Heeszel et al. (2016) would be more appropriate than the provided reference to Hansen et al. (2014).

How do the authors reconcile this? Simply writing off the difference between results to data sampling and crustal thickness and/or velocity variations in the seismic models (Lines 191-193) is a weak argument. Constraints from other geophysical studies, some of which are referenced in this manuscript, as well as others (e.g., Chaput et al., 2014; Ramirez et al., 2016) provide reliable crustal constraints for these models. A more robust comparison and analysis is warranted.

Lines 193-195: As worded, this sentence implies this model would be applicable along the full TAM front and not just within the study area. The authors have already stated that areas of the TAM closer to Ross Island and into Victoria Land are bounded by anomalously slow upper mantle. So extrapolating this model to the full TAM is likely an over-approximation.

Lines 200-203: It is true that the main phase of WA extension occurred at a different time (late Mesozoic) than the main phase of TAM uplift (middle-late Cenozoic), but it has also been argued that more recent WA extension is concentrated/localized along the TAM front, which the authors themselves point out earlier in the manuscript. So this time difference is likely not as significant as the authors indicate here.

Line 211-212: What is the lateral resolution off-profile (i.e., perpendicular to the MT transect)? The authors note that the RN feature in Figure 4 "grows toward Ross Island." How well is this feature resolved?

Lines 231-233: The low resistivities beneath the NE end of the MT profile (labeled MS on Figure 4) are interpreted to reflect large-scale metasedimentary bodies. This feature seems to be present at roughly 40-50 km depth, which given other geophysical studies of EA corresponds to the deepest part of the crust if not the upper mantle. Does this depth jive with the proposed interpretation? While a lesser component of the paper, this aspect could be better described.

Figure 1 and caption: It would be useful to label the glaciers mentioned in the caption (i.e., the Beardmore, Nimrod, Law, Walcott, Lennox, and King). Also, in the caption, a comma is needed on line 6, before "respectively."

Figure 2 and caption: As already noted, there are a variety of uplift mechanisms that have been proposed for the TAM, which the authors now illustrate (to some extent) here. However, no references are provided for the models shown in this figure. The authors need to include references to studies that proposed these mechanisms in the figure caption.

Figure 3 and caption: As someone who is not readily familiar with looking at MT images, this figure is very confusing. If the authors want to publish this work in a broader-readership journal like Nature Communications, they need to make their results more accessible to others outside their field. A reader should not have to decipher the SI to understand this – the paper should be able to stand on its own with the SI providing additional details for those who want this information. That being said, even with the SI, characteristics such as the impedance element phase (ϕ) are not well explained. I don't believe Re (plotted in rows 3 and 6 of Figure 3) is ever defined. Distill this down - what are the key characteristics of this image that are important to interpretation?

SI (in general): To be honest, as someone not familiar with MT analysis, much of the SI material didn't mean much to me. As I've already noted, I cannot provide a critical assessment of the methodology. That being said, it would have been helpful if critical parameters had been better described. For instance, in Figure 3, the authors provide plots of the impedance element phase (ϕ) and the parameter R_e , which I could not find a description for. I'm guessing this may stand for Resistivity, but then I'm unclear how this differs from ρ (the apparent resistivity).

SI, Line 176: The starting value for layers in the inversion was set to 50 ohm-m. If this is varied, are different results obtained? That is, are the inversion results dependent on this starting value?

Figure S1: The contrast of the lower picture isn't high enough to see the MT deployment well. Perhaps a higher resolution figure would make this more clear/obvious? Also, for photos like this (and those shown in Figure S2), it is generally required that credit is given to the photographer. This information should be added to the captions.

Figure S3: The colored legends on the upper right corners of these panels are very small and hard to read.

Figure S9: Again, the legend is very small and difficult to read.

Reviewer #3 (Remarks to the Author):

The authors present results from a 2D profile of MT data across the trans-Antarctic Mountains (TAM), and discuss implications for uplift of topography, which has been enigmatic. The MT dataset appears to be of very high quality, and all analysis, from data processing (with due consideration of possible source effects) to 3D inversion are state-of-the-art. The presentation is mostly very clear, and thorough – the supplemental material is very extensive (maybe too much so—the basic background on MT is probably not really needed, as there are plenty of references (which are given) to this material).

The interpretation seems reasonable and is mostly clear. I am not enough of an expert in this area to comment on whether the results are of very broad interest. Of the three models discussed for TAM uplift, the first (crustal root) is not really addressed by the MT results—but is perhaps already ruled out by other data. This is all stated clearly enough, but maybe slight rewording could emphasize that really this study can distinguish between the thermal and flexural hypotheses. (For example start with mechanisms 2 and 3, then just state that a thickened crust is another, but unlikely, possibility).

In the discussion it is stated that the MT results contrast with seismic evidence for low shear wave velocities below the TAM. I looked at the reference and I don't see this – for one thing the cited paper discusses only P-wave tomography! For another there are not low velocities beneath all of the TAM (at depths relevant to the MT results). Was there a different reference? Or is there really no clear disagreement with seismic results?

Overall I think this is a well written and interesting paper.

Reply to Reviewer #1:

We are quite grateful to the reviewer for his substantial efforts toward improving the substance and clarity of our presentation. We address the points according to original line number while also considering reviewer remarks in the returned pdf.

1. Done

25-27. We have modified the Abstract in an effort to reorder as the reviewer suggests and also meet journal guidelines.

38. "compositional" removed. We were actually referring to possible solid state density changes to the residuum as melt was removed but just refer to solid state density now. Also, regarding a comment in the pdf re line 37 qualifying African examples to East Africa, we also reference studies of the North African Hoggar Swell displaying late Cz mantle basalts from possible adiabatic decompression triggered by the Europe-Africa collision.

58-59. The original reference 13 is replaced with the recent Goodge and Fanning (2016) which supports our first two sentences.

92. The reviewer is correct to suggest we more explicitly mention the other data types that lead to essentially eliminating root buoyancy and thus allowing the MT results to more uniquely resolve the likely cause of uplift.

108-109. We have rewritten this sentence to say "currently inferred" to denote existing Te of 85 km, and that this is inferred not 'observed'. Similarly we emphasize that Te would reduce if radiogenic heating under the TAM or warming from WA occurred, according to the modeling done in former ref 33. Alternative non-thermal contributors to uplift apart from a crustal root are introduced here too and represented in Figure 2c with the enhanced lithospheric color contrast.

212. We specify that this is grid southeast.

224-230. We note that cratonic heat flow values should apply for the TAM and nearby polar plateau.

231-241. Based on laboratory experiments, expected high-grade hydrous minerals such as biotite or amphibole cannot be the cause of the very low deep crustal resistivities detected below the TAM and polar plateau as those do not appear to be more conductive than anhydrous silicates unless temperatures surpass ~500 C. Graphite or, somewhat less likely, sulfides are the favored explanation based upon other studies in similar stable regime settings as referenced. We note also that these bodies cannot represent shallower ice subcropping sedimentary units of the WSB.

General Comments. One cannot from simple visual inspection translate the raw observed data (former Fig 3) which are a function of EM field wave period to a tomographic (inversion) image of the subsurface (former Fig 4), so the disposition of color contours in the former must not be used to judge second order structures such as fault/boundary dip in the latter. It is a non-linear tomographic process involving fully 3D finite element simulations such as described with the former Supp Info Fig S9 (now Fig. 5). Although some of the contours may appear steep, those mainly represent the cascading effect of near surface structures onto the responses at longer

wave periods, which are unraveled from the top down through the inversion. Vertical, as opposed to dipping, contacts produce an apparent resistivity plunge on the conductive side that continues to amplify to long periods (e.g., ref 10).

Similarly, there is no innate tendency for the inverted resistivity structure to 'track' the MT stations. We presume the reviewer may be referring to the elongate conductor labeled MS in former Fig 4 near the northeast end of the profile. It is true that inversion images such as produced for MT (also seismic body and surface waves) are stabilized in the inversion by solving for the smoothest structure that can still fit the results so as to suppress artifacts. Also, longer period wave fields interrogate relatively larger average volumes so the images become smoother toward depth and laterally away from the data coverage. However, the narrow SW-NE linear behavior of MS is supported by the decrease in ρ_{yx} toward longer periods but with little influence upon ρ_{xy} . The narrow SE-NW arm near its middle is supported by the pronounced crossover in K_{zy} around 1000 s period near the NE end of profile (former Fig 3). Extending the series of model tests that we have presented could provide quantitative calculations to exemplify the prior remarks. However, since our main goal concerns the rift shoulder transition and thermal state below the TAM, the explicit test calculations concentrated upon those points to keep the (now) Methods section from being arbitrarily long.

Figure 1. SP and RI changed to white. WA and EA added in inset. Beardmore and Nimrod glaciers labeled BM and NR. RIS and PPI added. CTAM changed to orange like season 1 symbols, RISC changed to yellow like season 2. We also have shown the only passive seismic station within the field of view (MILR), with implications for seismic model resolution.

Figure 2. Color contrast is deepened to further distinguish the WA and EA lithospheres. However, we would rather not draw a T_e contour on the panels. As discussed in ref 33, the 85 km T_e value is somewhat of a model construct that does not necessarily strictly represent the viscous transition and can be affected by ice sheet/sediment presence and by erosion. T_e also will be reduced significantly in the case of thermal TAM support.

We have incorporated almost all the suggestions marked in the returned pdf. For the data in former Fig 3, we point out that $x=\text{grid315}$ is the x axis of the data quantities, and that the sections are for the central profile of stations along the transect.

Reply to Reviewer #2

We first wish to offer sincere thanks to the reviewer for the strong efforts made on behalf of improving the content and presentation of our submission. This confirms the importance of the study area and its implications for rift mechanisms here and worldwide. The reviewer feedback is structured initially as four main points (a-d) followed by specific comments. We thus will supply four overview responses followed by replies to the comments.

a) We have modified our discussion of the three principal hypotheses for rift shoulder uplift in a measured fashion. We did not intend for our paper to be a comprehensive review of the rift shoulder mechanism models that have been considered for the TAM, nor do we think that is appropriate. We present an MT geophysical data set with a model that is sufficient to point toward a preferred mechanism for uplift out of the most widely examined choices. The flexural cantilever model has been well-discussed in the literature on the TAM especially by ten Brink and by Yamasaki, which we based our succinct original discussion upon and referenced. We added material noting that in the cantilever model the surface deformation is entirely due to motion along the decoupling boundary fault with no other vertical loads applied. We have examined the other references offered by the reviewer and incorporate those as appropriate although we tended to opt for the most recent or central reference on any particular theme. We are allowed a maximum of 70, although we are begging an indulgence for 73.

b) We disagree that it was our intent to apply the conclusion that the cantilever model was most appropriate for the TAM as a whole. On the original line 204 we wrote "Physical state elsewhere along the TAM of course may vary.". Certainly along the more extensionally active Ross Is to North Victoria Land region, uplift mechanisms could well include thermal components and we indicated that in the original submission. We have strengthened the language on this point.

c) Indeed there is disagreement between our data for the central TAM and the seismic results of Heeszel et al (2016). As we in turn are not primarily expert in passive seismic methods, we do not wish to attempt to critique a particular seismic model in detail. We will note three things about Heeszel's model in our study area: 1, there is only one passive seismic station within the field of view of our Fig 1 (MILR), attesting to seismic data sparseness; 2, Heeszel et al state that "there is a tradeoff between upper mantle velocity and crustal thickness" so that "crustal thickness must be presumed a priori". Their crustal thickness estimates from a previous inversion changed little from initial guesses, perhaps due to insensitivity and regularization; and 3, their crustal thicknesses under the central TAM are less than those farther into East Antarctica whereas, as the reviewer notes, if anything Block's gravity suggests a slight crustal root beneath the central TAM where we examine it. We believe it is incumbent going forward that explicit tests of the seismic structure beneath the central TAM be made, because we have made such tests for our method in this submission. We also apologize for a reference misnumbering (25 should have been 26 in this section) which caused confusion.

d) We have worked to clarify the quantities shown in the former Fig 3 (now Fig. 4) and add some figures formerly in the Supp Info (now Methods) for illustration. These include an example apparent resistivity and impedance phase data sounding such as has been assembled into the data pseudosections. We also show the finite element mesh in the main paper section to illustrate how the 3D model was constructed. However, we believe it is the appropriate role of the Methods to carry detailed descriptions of the quantitative analytical techniques underpinning the models presented and discussed in the main paper. The additional reviewer who appeared to be an MT expert endorsed our methodology.

Specific Comments by original line number:

33. Done.

39-43. We have qualified that statement although we maintain that thermal causes are at the forefront of consideration in rift shoulder support for continental interiors especially in recent years. This is the case both for Antarctica and worldwide, with a minority of exceptions such as possibly Lake Baikal. Fitzgerald et al (1986) considered lithosphere-asthenosphere conversion and magmatic underplating as the uplift cause (thermal uplift case). Lawrence et al (2006) suggest at least some portion of TAM uplift is associated with buoyant thermal load in the mantle beneath the edge of TAM lithosphere (thermal uplift case) as they recognized no sufficient crustal root exists. Chery (1992) interpret that TAM (rift flank) uplift results from flexural bending of the lithosphere resulting from an isostatic load following lithospheric necking, but van der Beek et al (1994) point out that does not fit known gravity or crustal structure and argued for lithospheric thinning under the TAM (thermal uplift case) possibly by simple shear. For conciseness, we focused on the most recent work emphasizing geodynamic modeling plus the most recent (at submission) seismic models. Similarly we had considered Stern and ten Brink (1989) to be contained within ten Brink et al (1997), and Studinger et al (2004) within Block et al (2009) for conciseness, but have expanded the ms to include these in the revision. We certainly considered the other geodynamic contributions, but if non-thermal causes dominate across the central TAM, then they would warrant a reexamination elsewhere along the TAM.

49. Corrected.

59. Word "to" is replaced by "against".

64-65. Uplift time specified and Fitzgerald (2002) referenced again.

70. Noted.

74. Done.

75-78. We restate it as "relatively coarsely sampled" but do not agree that adequate seismic coverage to address uplift exists for the central TAM. For instance, there is only one seismic station (MILR) within the field of view of the main panel of Figure 1. The Polenet profile passes far to the grid west, and west of the South Pole. The grid westernmost Tamseis stations end at Byrd glacier, well outside our study also. We agree that coverage and resolution from Ross Island geographic northward to Northern Victoria Land is much better, suggesting a thermal contribution to uplift, and recognized so on line 210 in our original submission. We emphasize this further in the revision. That still does not negate a possibly strong mechanical contribution to uplift along the TAM.

92. Sentence rewritten.

80-81. Expanded as appropriate.

84-93. There is nothing inconsistent with referencing the subsidence and apparent lithospheric necking occurring under the Ross Sea margin as illustrated from independent sources by Huerta and Harry (2007) (orig line 67), and subsequently ruling out the importance of a crustal root in TAM uplift based largely on later seismic receiver function work (e.g., Hansen et al., 2009).

These are two different matters. The concentrated lithospheric stretching concept supports the subdued version of such that we image under our MT transect. Huerta and Harry in part entertained a root to illustrate their concept of crustal heat production contributing to a thermal load, which I think most everyone agrees now doesn't play strongly.

Regarding Block et al., as we pointed out, the thickest possible TAM root from the GRACE data lies tangentially proximal to South Pole under Neoproterozoic basement distinct from that of the central TAM. Moving grid SE to the Miller Range near our study, the inferred extra thickness has diminished to a negligible 1-2 km in their Figure 3. Thus, a significant crustal thickness component to uplift in our study area can be reasonably ruled out.

97. Added.

99-101. We inadvertently misnumbered the reference superscript here, that's all. Apologies. The original ref 25 (Hansen) should have been 26 (Heeszel). The latter of course considers seismic structure beneath the entire length of the TAM including our seismically sparsely sampled study area. Above in our general reply we disagree that the evidence for slow uppermost mantle in our study area is good, given the direct trade-off between crustal thickness constrained a priori and upper mantle model velocity. We maintain that it should be the seismic model which should be explicitly tested here in this regard, because we have done such testing with our data.

101-104. The comparison of the WA-EA transition with the margins of the Great Basin has been made by others that we cite. This is an appropriate place to bring it up because we will be comparing geophysical structures of the two regions in our tests of these dynamic hypotheses.

105-110.

a) Yes, the uplift is variable and influenced by several factors. The value 5 km is within the range considered by several researchers (e.g. Yamasaki et al., 2008), is compatible with fission track data in our study region (4-6 km; Stern et al., 2005), is compatible with recent estimates over the nearby Shackleton Glacier area (4.9-6 km; Miller et al., 2010) and is a sufficient working value for the discussion at hand. The precise value is not critical for our purpose, which primarily is to demonstrate that the central TAM is not underlain by thermally elevated lithosphere.

b) Correct, it is a modeled value and we modify the sentence.

c) We would not agree that the T_e estimates depend "greatly" upon assumed uplift (ie., 85 vs 115 km). T_e is in the neighborhood of 100 km.

d) The reviewer appears to disregard the results of Yamasaki et al. (2008) which we consider. Their approach utilized a rigorous finite element simulation accounting for realistic rheologies as well as contrasts in crustal and mantle lithospheric thickness between EA and WA, and heat production. Earlier simulations such as by Stern and ten Brink (1989), while illuminating, relied upon idealized broken plate models and forcing functions. Moreover, in offering the cantilevered flexure model as preferred, it is sufficient to show that discussed alternatives are significantly less likely. In particular, we wouldn't be evaluating thermal load effects in our study area because our data indicate that there are not significant ones in the sense that they have been discussed elsewhere along the TAM.

119-120. In the 12 years between the South Pole field deployment and that at CTAM, modest improvements in electronic components and layout for field instrumentation can be expected. The amplifier design merely was simplified and made more robust, but it is still a buffer amplifier that breaks the high contact impedance at the firm and presents a normal-scale electronic output impedance to the central recorder. The buffer amplifier concept has been in the public domain

since the middle 1970s and readers can follow the literature if they are interested in details. There are not to date other MT experiments along the TAM front.

126. As discussed and tested in the Supp Info (now Methods), our source wavefields penetrate to the ~410 km phase transition broadly taken as the lower limit of the upper mantle. We now mention in the 3D inversion section (new line 199) that resolution versus depth (or distance) from the observation points decreases in a scaled fashion because greater distances are reached by lower frequency (longer period) source wave fields which interrogate relatively larger earth volumes. We make sure that the elements in the finite element mesh are smaller than this resolving scale.

127-132. Pseudosections are simply a summary representation of the total MT data set in compact form, and have been employed for decades. Some of their gross characteristics can be used to illustrate heuristically first order structural influences on the measurements. Data displays have nothing directly to do with the resolution of the method, which must be assessed through rigorous non-linear inversion and feature testing as we have done with the deformable finite element method. We have moved some of the material of the Supp Info (now Methods) to the main text hopefully to provide a better feeling for the method.

154-155. This appears to be a misreading of our model construction and testing on this point. As discussed, an abrupt decrease in electrical resistivity across the ~410 phase transition is well founded from laboratory measurements and supported by field studies that extend to greater source wave periods than ours (orig ref 37). Thus we impose it in our a priori/starting model, and the required departures from it across the 410 km depth range are minimal to agree with our data. However, if it is removed and the upper starting value of 300 ohm-m is fixed at depths greater than 410 km, then an extreme and unlikely (from an Occam standpoint) band of low resistivity with a poorer data fit is the result just above the 410. This indicates that our data have some sensitivity to this depth range although this region is not central to the main thrust of the submission regarding thickness of the cold TAM lithosphere and its implications for TAM geodynamics.

158-160. The width chosen largely reflects the width of the reduced uppermost mantle velocities shown in the 60-120 km depth range in Heeszel et al. (2016), which is ~200 km. In the updated model of ten Brink and Stern (1997), where new data of the EAST93 transect were incorporated and the decoupling of TAM response from Ross Sea recognized, the thermal load was distributed over a 100 km width. However, we are certain that the results of the test would change little because by far the most resistive TAM lithosphere is that closest to the range front. This is reflected in the degree of induced data misfit as represented by the nRMS site diagram in former Figure S17 (now Methods). The second part of the query is somewhat a tautology. A neighboring region affected by a source of heat is by definition heated. Heating upper mantle below the TAM to the point of being anomalous provides the buoyancy load while at the same time changing the electrical geophysical properties to an extent that our measurements would resolve. Transforming only the upper mantle in this fashion is actually a conservative change because both rift shoulders of e.g. the US Great Basin exhibit magmatic underplating responses that are much stronger than that which we already overrule with MT.

173-174. Indeed, the ability to perform 3D inversion of quasi-profile (but including some off-profile sites) is a strength. The full tensor nature of the acquired data including vertical magnetic field means that it has off-line and directional sensitivity. Granted, resolution diminishes with distance from the profile just as it does with depth. However the inversion explains the entire data set, which 2D inversion is far from guaranteed to do as former Figures 3 and S14

demonstrate. The 3D inversion regularization produces a model which is as smooth as possible while still fitting the data so the offline structures can be considered conservative representations of actual resistivity structure. The model provides a useful distribution estimate for likely major metasedimentary packages in the vicinity of our transect.

S177. “normalized root-mean-square” added.

179-180. Once more, apologies for the reference number mixup where former ref 25 should have been 26. As for the differing seismic and MT results, we outlined our issues with the seismic model in this region in our general response above c). We do not wish to be strident in how this is handled in our text. We add that the permissible range of crustal thickness from the seismic data was not specifically tested in the central TAM area to establish the effect of the tradeoff. The Chaput paper concentrated upon WA, and the isolated Miller Range site near our study gave a crustal thickness of 45 km +/- 10 km. Ramirez did not even consider the Miller Range site. The a priori crustal thickness used by Heeszel along the central TAM appears mostly to be <40 km. We believe the seismic model is the one which requires more explicit testing such as we have done in this submission with our model.

193-195. We specify central TAM more clearly.

200-203. Although continued extension particularly along the Terror Rift zone is recognized by all, this does not have similar surface or seismic expression through the central TAM.

211-212. This particular feature is well resolved and determined by the strong positive anomaly in $\text{Real}(K_{zx})$ at long periods seen in former Figure 3. We expand upon data features that determine certain structures.

231-233. The transport of possibly graphitized metasedimentary material to the deep crust during terrane suturing has been inferred in many orogenic scale MT studies. These are reviewed in former ref 45 plus another reference added. They appear in the recent Earthscope paper by Murphy and Egbert on the Southern Appalachians.

Figure 1. Labels are added for Beardmore and Nimrod, but would be too crowded to add others right under the transect. Also WA, EA, RIS and PPI added. Comma added ahead of ‘respectively’.

Figure 2. References behind these hypotheses are presented in the main text, which is now noted in the caption. We believe it is redundant to recite.

Figure 3. We have expanded this section by moving material from the former Supp Info to be explanatory. Pseudosections are a summary data display where only gross characteristics should be viewed as indicating a few major structural or property characteristics. Omitted symbol definitions are corrected. We defined the MT quantities now in the main text. ‘Re’ is standard notation for real component of a complex quantity. Notions of apparent resistivity and impedance phase are the most basic MT definitions; they are commonplace in the literature including the recent comprehensive textbook by Chave and Jones¹⁰.

S176. The starting model should be as simple as possible so as to not be prejudicial, but it is appropriate that certain aspects of the data or some well founded constraints be considered. Therefore, the half-space starting value is close to the average resistivity of the entire model domain based upon an average (integration) of the observed MT impedance data over all

stations and all wave periods. This reduces “strain” in the inversion where fitting the data subject to minimizing model slope in three dimensions is sought. A starting model such as this is analogous to seismic inversions that start from the PREM model. Heeszel et al in fact use separate 1D average starting models for WA and for EA, which is a more complex starting model than we utilize and surely has some residual effect across the TAM.

Figure S1. Unfortunately, that is the best whole-site photo we have. We enlarged it for the Supp Info and the electric bipole lines are visible upon close inspection. Photo credits added to captions.

Figure S3 (now S5). We have enlarged these panels to 150% of previous.

Figure S9. These we have enlarged.

Reply to Reviewer #3:

We thank the reviewer for examining and verifying our procedure in collection and analysis of MT data in this Antarctic setting. We believe it is important to be thorough in the approach to ensuring high-quality, plane wave equivalent responses as this is a common concern in high-latitude studies. Geoscientists outside of MT commonly ask about MT resolving capability and so thorough tests of model features critical to tectonic inferences are essential. The opening MT introduction in the Supp Info (now Methods) amounts to a small fraction of total length so we retain it to briefly define terms.

The ordering of hypotheses for mechanisms of TAM uplift is somewhat a style preference. The crustal root possibility has been widely applied in other settings and received substantial attention at the TAM for some years until further data appeared to clarify the matter, so we wished to dispense with it first.

Regarding the contrast between the MT and seismic tomography results, we inadvertently numbered the reference incorrectly, where 25 should have been 26 in the original submission. With the latter (Heeszel), surface wave data were used to infer a band of reduced Vs in the uppermost mantle along the length of the TAM. In the former reference, only P-waves were utilized as the reviewer correctly notes, and together with the sparse station coverage a low velocity volume near our study area was not resolved. So it is the results of Heeszel with which our results are at odds. However, we have explicitly tested the possibility of a thermally enhanced upper mantle and found it to be inconsistent with our data.

REVIEWERS' COMMENTS:

Reviewer #1 (Remarks to the Author):

Review of revised manuscript

I am pleased to see that this manuscript has been revised and resubmitted. I think it is a very valuable study and will be of interest to not only the Antarctic geoscience community, but to a broader audience worldwide interested in geodynamic and tectonic controls on uplift and its relation to lithospheric structure in a generally extensional regime. I recommend that it be accepted for publication.

1. I have read the authors' response to the three reviewer comments, and I think they have thoroughly addressed all of the comments, especially the more substantive reviews #1 and #2. In my view they show they are well aware of the criticisms and any shortcomings highlighted by the reviews and have addressed them very adequately either in the revision or the reply.

2. I have perused the revised manuscript and figures, and the changes made by the authors all look good. Overall, I think the manuscript reads well and is very accessible even though compact. The figures are in good shape and helped by the changes made.

One minor tweak to the abstract, line 30: It is not really appropriate to say "is a more applicable mechanism..." if not followed by something else. More than what? No other mechanism is defined. I suggest revising to read "is the most valid support mechanism..." or something along those lines. If the "more applicable" refers to "negligible thermal encroachment" on line 27, that is really not stated as a possible mechanism and when prefaced by negligible does not convey it as a valid alternative. For purposes of the abstract, I think it is best to keep it focused on the data/properties reported for the lithosphere in the two areas (EA and WA) and posit a reasonable mechanism, thereby avoiding confusion by raising alternatives.

Reply to REVIEWERS' COMMENTS:

Reviewer #1 (Remarks to the Author):

Review of revised manuscript

I am pleased to see that this manuscript has been revised and resubmitted. I think it is a very valuable study and will be of interest to not only the Antarctic geoscience community, but to a broader audience worldwide interested in geodynamic and tectonic controls on uplift and its relation to lithospheric structure in a generally extensional regime. I recommend that it be accepted for publication.

1. I have read the authors' response to the three reviewer comments, and I think they have thoroughly addressed all of the comments, especially the more substantive reviews #1 and #2. In my view they show they are well aware of the criticisms and any shortcomings highlighted by the reviews and have addressed them very adequately either in the revision or the reply.

2. I have perused the revised manuscript and figures, and the changes made by the authors all look good. Overall, I think the manuscript reads well and is very accessible even though compact. The figures are in good shape and helped by the changes made.

One minor tweak to the abstract, line 30: It is not really appropriate to say "is a more applicable mechanism..." if not followed by something else. More than what? No other mechanism is defined. I suggest revising to read "is the most valid support mechanism..." or something along those lines. If the "more applicable" refers to "negligible thermal encroachment" on line 27, that is really not stated as a possible mechanism and when prefaced by negligible does not convey it as a valid alternative. For purposes of the abstract, I think it is best to keep it focused on the data/properties reported for the lithosphere in the two areas (EA and WA) and posit a reasonable mechanism, thereby avoiding confusion by raising alternatives.

We have reworded the abstract based upon this and Associate Editor feedback so that this phrase no longer appears in the abstract.

PW

** See Nature Research's author and referees' website at www.nature.com/authors for information about policies, services and author benefits

This email has been sent through the Springer Nature Tracking System NY-610A-NPG&MTS